



# Ensemble quantification of short-term predictability of the ocean dynamics at kilometric-scale resolution: A Western Mediterranean test-case.

Stephanie Leroux[1,2], Jean-Michel Brankart[1], Aurelie Albert[1,2], Laurent Brodeau[1,2], Jean-Marc Molines[1], Quentin Jamet[1], Julien Le Sommer[1], Thierry Penduff[1], and Pierre Brasseur[1]

[1]Univ. Grenoble Alpes, CNRS, IRD, Grenoble INP, IGE, 38000 Grenoble, France
[2]Ocean Next, Grenoble, France

**Correspondence:** Stephanie Leroux (stephanie.leroux@univ-grenoble-alpes.fr)

**Abstract.** We investigate the predictability properties of the ocean dynamics using an ensemble of short-term numerical regional ocean forecasts forced by prescribed atmospheric conditions. In that purpose, we developed a kilometric-scale, regional model for the Western Mediterranean sea (MEDWEST60, at 1/60° horizontal resolution). A probabilistic approach is then followed, where a stochastic parameterization of model uncertainties is introduced in this model to initialize ensemble predictability experiments. A set of three ensemble experiments (20 members and 2 months) are performed, one with the deterministic model initiated with perturbed initial conditions, and two with the stochastic model, for two different amplitudes of stochastic model perturbations. In all three experiments, the spread of the ensemble is shown to emerge from the small scales (10 km wavelength) and progressively upscales to the largest structures. After two months, the ensemble variance saturates over most of the spectrum, and the small scales (<100 km) have become fully decorrelated across the ensemble members. These ensemble simulations are thus appropriate to provide a statistical description of the dependence between initial accuracy and forecast accuracy for time-lags between 1 and 20 days.

The predictability properties are statistically assessed using a cross-validation algorithm (i.e. using alternatively each ensemble member as the reference truth and the remaining 19 members as the ensemble forecast) together with a given score to characterize the initial and forecast accuracy. From the joint distribution of initial and final scores, it is then possible to quantify the probability distribution of the forecast score given the initial score, or reciprocally to derive conditions on the initial accuracy to obtain a target forecast skill.The misfit between ensemble members is quantified in terms of overall accuracy (CRPS score), geographical position of the ocean structures (location score), and spatial spectral decorrelation of the Sea Surface Height 2-D fields (decorrelation score). With this approach, we estimate for example that, in the region and period of interest, the initial location accuracy required (necessary condition) with a perfect model (no model uncertainty) to obtain a location accuracy of the forecast of 10 km with a 95% confidence is about 8 km for a 1-day forecast, 4 km for a 5-day forecast, 1.5 km for a 10-day forecast, and this requirement cannot be met with a 15-day or longer forecast.





# 1 Introduction

The Copernicus Marine Environment Monitoring Service (CMEMS) routinely provides analyses and forecasts of the state of the ocean, to serve a wide range of marine scientific and operational applications. Most CMEMS systems rely on the NEMO (Nucleus for European Modelling of the Ocean) modelling framework to embed state-of-the-art representations of the various dynamical components of the ocean, with the goal to improve the accuracy and the resolution of the products. However, with the increase of the complexity and resolution of the model, new questions arise regarding the predictability of the system. To what extent is it possible (and does it make sense) to forecast the very fine scales (∼kilometric) targeted by the future generations of CMEMS systems using the NEMO dynamical core? How is this forecast sensitive to initial errors or to possible shortcomings or approximations in the model dynamics? These questions are important for CMEMS and other operational centers because they can help rationalizing expectations from the next systems and thus help driving future developments.

Historically, the question of the predictability of dynamical systems has been addressed by considering only the irreducible sources of error, which result from intrinsic model instability combined to inevitable small initial errors. In a deterministic framework, modelling errors can indeed be excluded from the analysis because they can be reduced by additional modelling efforts, so that they do not represent a theoretical limitation to predictability. There is a long history of studies along this line, starting with Lyapunov (1992), who suggested looking for the fastest-growing unstable modes (Lyapunov vectors) and their associated e-folding timescales (Lyapunov exponents). This was extended in meteorology to describe the largest error growth over a finite time (with singular vectors, Lorenz, 1965; Lacarra and Talagrand, 1988; Diaconescu and Laprise, 2012), before it was recognized that linear instability studies were quite often not sufficient to provide a correct picture of the predictability patterns, even for quite short time lags. Nonlinear model integrations are needed to allow the fast instabilities to saturate, and reveal the patterns that really matter over a given forecast time (e.g. Lorenz, 1982; Brasseur et al. , 1996). For this reason, the bred vectors (Toth and Kalnay, 1993; Kalnay, 2003) have been introduced as a practical way to identify the most relevant perturbations to initialize ensemble forecasting systems. In the meantime, ensemble forecast simulations, explicitly performed with the full nonlinear model, have become the standard approach to investigate predictability (e.g. Palmer and Hagedorn, 2006; Hawkins et al., 2016). Performing an ensemble forecast amounts to propagating a probability distribution in time, which includes the possibility of a non-deterministic model. In this framework, it is thus possible to go beyond the assumption that predictability is mainly limited by unstable and chaotic behaviours, and to include the possibility that model uncertainties can be an essential limiting factor to forecast accuracy, as also recognized recently in the work of Juricke et al. (2018).

In the last two decades, indeed, more and more studies have suggested that uncertainties are intrinsic to atmosphere and ocean models, as long as they do not resolve the full diversity of processes and scales at work in the system (e.g. Palmer et al. , 2005; Frederiksen et al., 2012; Brankart et al., 2015). Non-deterministic modelling frameworks have been shown very helpful to improve the accuracy of medium-range weather forecasts (Buizza et al. , 1999; Leutbecher et al., 2017), to enhance their economical value (Palmer, 2002), to alleviate persistent biases in model simulations (Berner et al., 2012; Juricke et al. , 2013; Brankart, 2013; Williams et al., 2016), and to account for some misfit between model and observations in data assimilation systems (e.g. Evensen, 1994; Sakov et al., 2012; Candille et al., 2015). In any case, whether the system can be thought as





fundamentally deterministic or not, in practice all CMEMS systems involve substantial modelling uncertainties. What matters to the application is then the possibility to produce a valuable forecast with the model that is used, and which may be quite different from what could be obtained by only considering the unstable or chaotic behaviour of a perfect deterministic model.

For these reasons, our objective in this study is to evaluate, in practice, the predictability of the ocean fine scales in a high-resolution (kilometric scale) NEMO-based model, to anticipate the future generations of CMEMS systems that will aim for such resolution. Both the effect of initial uncertainties and model uncertainties are considered, either separately or together. In both cases, it will be assumed that they cannot be made arbitrarily small in CMEMS systems: initial uncertainties because observation resources are limited, and model uncertainties because model resources are limited. Nevertheless, these finite-size uncertainties may have very different origins and may display very different shapes in space and time, so that an assumption is still needed to simplify the problem. In this study, this simplification will be obtained by considering one generic type of model uncertainty that primarily affects the small scales of the system. By tuning the amplitude of the perturbations, we can then simulate different levels of model accuracy, and generate ensemble initial conditions with different levels of initial spread. With this assumption, we can then quantify the accuracy of the forecast that is obtained, for a given combination of initial and model uncertainties.

And reciprocally, we can then expect that this set of experiments can provide insight on the maximum level of initial and model uncertainties that is required to obtain a given forecast accuracy. This might give an idea of the relative importance of the initial and model uncertainties to obtain an accurate forecast of the small scales, and thus the relative weight of the observation and model constraints in the quality of the CMEMS- products, and maybe help us understand the level of initial and model accuracy required to produce a useful forecast of the small scales, as targeted in the future kilometric-scale operational systems. However, it is important to keep in mind that these conclusions will depend on the assumption made to simulate uncertainties in the system. Although generic, and designed to trigger perturbations in the small scales, they are still an approximation and cannot be expected to account for the full diversity of uncertainties of different kinds propagating in real operational systems.

In section 2, we present the kilometric-scale regional model based on NEMO over the Western Mediterranean sea that we set up for this study. We then introduce the parametrization for model uncertainties that is used to generate different levels of initial spread and model accuracy. In section 3 we present the three ensemble experiments produced with these settings, and we assess and compare their spread growth. Predictability diagnostics are then illustrated in section 4 by applying different types of metrics (probability scores, location errors, spectral analysis) to characterize the dependence of the forecast accuracy to initial and model uncertainties. We finally summarize the outcomes of this study in section 5.

## 2   A kilometric-scale regional ocean model

### 2.1   Model specifications and spin-up

A kilometric-scale regional model of the ocean circulation based on NEMO has been developed for the Western Mediterranean sea, using boundary conditions from a larger reference simulation (covering the entire North Atlantic) at same resolution (eNATL60, Brodeau et al., 2020). The new regional model, MEDWEST60, covers a domain of 1200 km × 1100 km, from





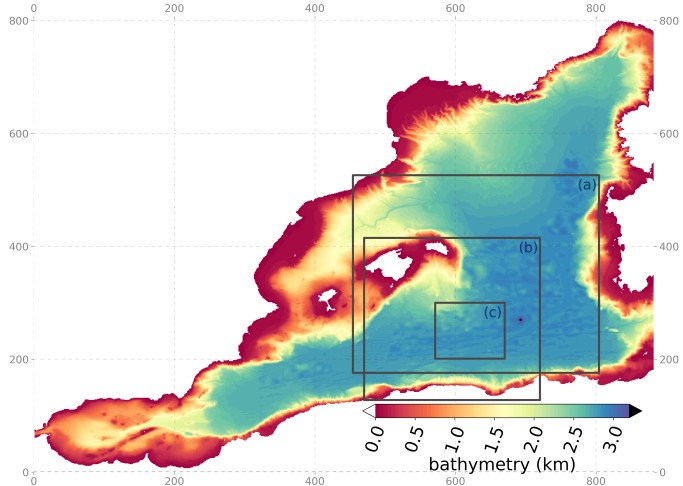

**Figure 1.** Bathymetry (in km) of the MEDWEST60 regional domain (x- and y- axes are model grid points). The full domain covers 883 x 803 grid points in the horizontal, representing 1200 km x 1100 km, from 35.1° N to 44.4° N in latitude and from 5.7° W to 9.5° E in longitude. The black squares localize three subregions which are refered to in the text.

35.1° N to 44.4° N in latitude and from 5.7° W to 9.5° E in longitude (see Figure 1). This configuration includes explicit tidal

motion (tidal potential), and is forced at the western and eastern boundaries with hourly outputs from the reference simulation eNATL60 (which also includes tides). By design, all parameter choices for MEDWEST60 were made with the idea to remain as close as possible from the reference simulation eNATL60. The MEDWEST60 specifications are summarized in Table 1. We use strictly the same horizontal and vertical grids as eNATL60, meaning that there is no need for spatial interpolation of the lateral boundary conditions. Besides the smaller domain, the only differences between MEDWEST60 and eNATL60 are:

(a) the lateral boundary conditions, (b) the model time-step, increased by a factor 2 (80 seconds in MEDWEST60 versus 40 seconds in eNATL60), and (c) there is no additional tidal harmonic forcing at the lateral boundaries in MEDWEST60 since the tidal forcing at the boundaries is already explicitly part of the hourly boundary forcing from eNATL60 outputs.

By design, the MEDWEST60 model can be initiated with an instantaneous, balanced 3-D ocean state archived from the reference simulation eNATL60 on the same horizontal and vertical grids. Our spinup protocol is as follows: from a NEMO

restart file archived from eNATL60 on a given date (here 25th Jan. 2010), we extract the horizontal and vertical domain corresponding to MEDWEST60. A first regional simulation is then run for 5 days, started from the extracted restart file, and using the same time-step as eNATL60 (i.e., $\delta t$=40 seconds). Five more days are then run with a doubled time-step of $\delta t$=80 seconds, and a new MEDWEST60 restart file is finally archived, to be used as the starting point on the 5th Feb. 2010 for the following ensemble forecast experiments.





| | |
|---|---|
| Numerical code: | NEMO 3.6 |
| Horizontal resolution: | $1/60°$ |
| Grid size: | 883 x 803 in the horizontal<br>($1.20$ km $<\Delta$x$<1.55$ km) |
| Vertical grid: | 212 levels (same as eNATL60)[1] |
| Timestep: | 80 s |
| Atmospheric forcing: | 3-hourly ERA-interim (ECMWF) |
| Tidal potential | On |
| Lateral boundary conditions<br>(ocean): | 1-hourly eNATL60 simulation<br>(including tides) |
| Lateral boundary conditions<br>at the coast: | No slip |

**Table 1.** Specifications of the MEDWEST60 model.

(1) The vertical levels are defined exactly as in eNATL60 but only 212 levels are
actually needed to include the deepest points in the Western Mediterranean region (i.e
3217 m at the deepest), while 300 levels were used in eNATL60 to cover the depth
range in the North Atlantic basin. The following discretisation is applied to the first 20
meters below the surface: 0.48 m, 1.56 m, 2.79 m, 4.19 m, 5.74 m, 7.45 m, 9.32 m,
11.35 m, 13.54 m, 15.89 m, 18.40 m, 21.07 m.

## 2.2 Paramerization of model uncertainties

The model presented above is a deterministic model, in the sense that the future evolution of the system is fully determined by the specification of the initial conditions, the boundary conditions and the forcing functions. This type of model - deterministic - is the archetype of the models that are currently used in CMEMS operational forecasting systems (though not yet at kilometric scale). In this context, forecast uncertainties can only be explained by initial uncertainties, boundary uncertainties or forcing uncertainties, usually amplified by unstable model dynamics. However, as motivated in the introduction, the objective of this study is to go beyond this assumption and include the possibility of model errors impairing the predictability of the finest scales.

We thus transform the deterministic model presented above into a stochastic model, with the ambition to emulate uncertainties that primarily affect the smallest scales of the ocean flow, and let them upscale to larger scales according to the model dynamics. These uncertainties are likely to depend mostly on the intimate structure of the model, by embedding misrepresentations of the unresolved scales and approximations in the numerics of the model. A detailed causal examination of the origin and interactions between these various possible sources of error being quite impossible to achieve, we propose to introduce





here a bulk parameterization of these effects, by assuming that one of the most important dynamical consequence of these errors on the finest scales is to generate uncertainty in the location of the oceanic structures (currents, fronts, filaments,...).

In fluid mechanics, there is an ample literature explaining that the effect of unresolved scales in a turbulent flow can be described by uncertainties in the location of the fluid parcels (e.g., Griffa, 1996; Berloff and McWilliams, 2002; Ying et al., 2019). This general idea is applied for instance in the work of Mémin (2014) and Chapron et al. (2018), where the Navier-Stokes equations are modified by adding a random component to the Lagrangian displacement **dX** of the fluid parcels (as in a Brownian motion). In the present study, location uncertainties are introduced in our ocean model according to a very

similar approach, except that the random perturbations are directly applied to the discrete model (rather than the mathematical equations), in the form of stochastic fluctuations of the horizontal numerical grid. A more detailed description and justification of this parameterization is provided in appendix A. In summary, the effect of the parameterization is to perturb the horizontal metrics of the model (i.e. the size of the horizontal grid cells) using a multiplicative noise with specified time and space correlation structure. In our application, the correlation scales of this noise have been set to 1 day and 10 grid points, to be

smooth enough and nonetheless produce perturbations on the small-scale side of the spectrum. The standard deviation of the noise can also be tuned, so that it is easily possible to simulate different levels of model accuracy, as required in our experiments to generate different levels of initial ensemble spread (see section 3). This standard deviation must however remain small with respect to the size of the model grid cells so that the perturbations that do not impair the physics of the model for the resolved scales. In practice, we have used values of 1% and 5%.

The two main effects that this parameterization is expected to produce in the model are on the horizontal advection and on the horizontal pressure gradient. In the advection scheme, the stochastic part of the displacement **dX** of the fluid parcels is directly accounted for by the displacement of the grid, and in return, the transformed grid induces modifications in the advection by the resolved scales. On the other hand, location uncertainties also produce fluctuations of the horizontal pressure gradient (by shifting the position of the tracer fields), which is quite consistent with the conclusions obtained by Mémin (2014). It is

therefore expected that these stochastic fluctuations can bring a substantial limitation to the predictability of the small-scale motions. Overall, we can see indeed that non-deterministic effects are produced in several key components of the model, and that all of these effects consistently derive from the sole assumption that the updated location of the fluid parcels after a model time step is not exact, but approximate.

## 3    A set of 3 ensemble forecast experiments

Three ensemble forecast experiments were performed with MEDWEST60, to investigate predictability as a function of both initial uncertainty and model uncertainty. In this section, we give a description of these three ensemble experiments, and how they were initialized. We then assess how the spread grows with time in those ensembles, comparing the results from the probabilistic model (with model uncertainty) and from the deterministic model (no model uncertainty). The predictability diagnostics will be presented in section 4.





## 3.1 Generating the ensembles

Two experiments (ENS-1% and ENS-5%) are performed with the probabilistic model (i.e. including model uncertainty) and starting from the same perfect initial conditions on the 5th Feb. 2010. Those two ensemble experiments explore two different amplitudes of the stochastic scheme described in section 2.2 and Appendix A. Experiments ENS-1% and ENS-5% are set for a stochastic perturbation of standard deviation 1% and 5% of the horizontal grid size, respectively (see illustration in Fig. 2). By design, the other parameters of the stochastic module are kept identical in all the experiments: the time correlation is set to 1 day (1080 timesteps), and the laplacian filter introducing spatial correlations is applied 10 times.

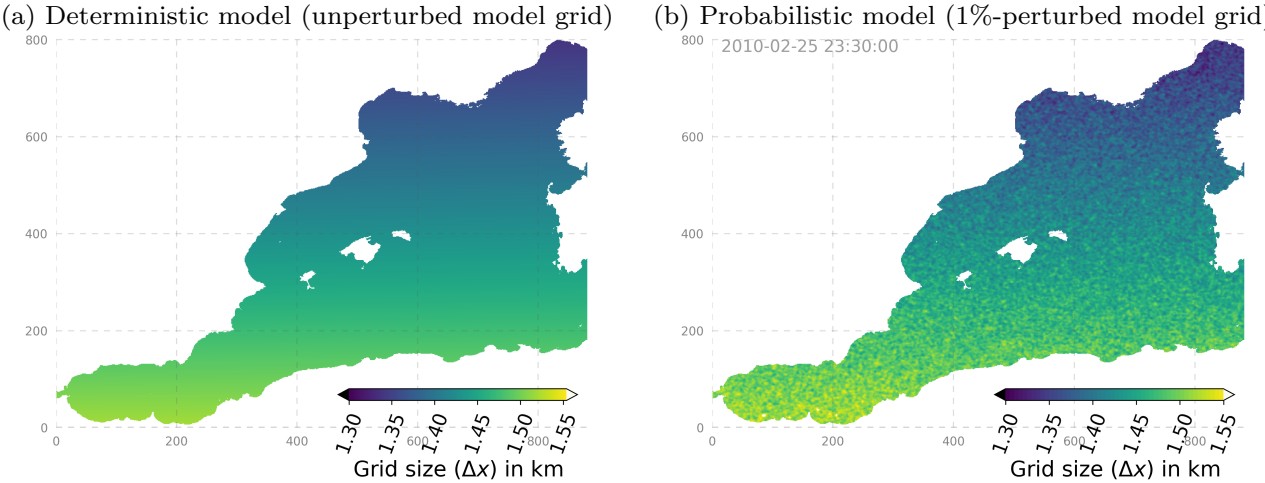

**Figure 2.** Size of the model grid in the horizontal east-west dimension (`e1t` in NEMO): (a) unperturbed, from the standard NEMO grid at 1/60° resolution, and (b) snapshot of the perturbed metric at a given date (stochastic perturbation set to a level of std=1%).

The third ensemble experiment performed (ENS-CI) is the experiment with the deterministic model (i.e. no model uncertainty) to study predictability under imperfect initial conditions. It is initialized from ensemble conditions taken from experiment ENS-1% after 1 day of simulation (i.e. on the 6th Feb, 2010) when the states of the 20 members have already slightly diverged on the fine scales. Note that the choice is made to start experiment ENS-CI with small initial errors, but this experiment also virtually gives access to forecasts initialized with larger errors by considering day 1, day 2, (...) day 10, etc, of ENS-CI as many different start times. This approach will be applied in the predictability diagnostics proposed in the next section (section 4). It does rely on the assumption that changing the initial date of the ensemble forecast of a few days does not influence significantly the predictability results. An alternative approach would have required performing a large number of ensemble forecasts with various levels of uncertainty on the initial conditions (for a same start time), and would have been much too expensive in computing ressources. Following the same idea, experiments ENS-1% and ENS-5% also virtually give access to forecasts accounting *both* for model error and some initial error by considering day 1, day 2, day 10, etc, of the experiments as many different virtual start times with increasing initial error.

Table 2 offers a summary of the three ensemble forecast experiments and their characteristics.




| Experiment | ENS-1% | ENS-5% | ENS-CI |
|---|---|---|---|
| Start date: | 05-02-2010 | 05-02-2010 | 06-02-2010 |
| Length (in days): | 60 | 60 | 60 |
| Ensembe size: | 20 | 20 | 20 |
| Initial cond.: | identical | identical | perturbed[1] |
| Restart from: | spinup | spinup | day 1 of ENS-1% |
| Model type: | probabilistic | probabilistic | deterministic |
| Stochastic param.: | e1,e2 | e1,e2 | *none* |
| & amplitude: | std=1% | std=5% | - |

**Table 2.** Characteristics of the three ensemble forecast experiments ENS-1%, ENS-5%, ENS-CI. (1) The "perturbed" ensemble initial conditions of experiment ENS-CI are taken from the restart files of experiment ENS-1% after 1 day of simulation (see text).

## 3.2 Impact of the location uncertainty on the model solution

In the section below we assess how the spread grows with time in those three ensembles. As explained in Appendix A, the stochastic perturbation is applied on the model horizontal metrics, while the location of the grid points themselves is assumed to be the same for all members. In other words, the field itself is still considered to be located on the reference grid, for instance with respect to the bathymetry and the external forcing, and the effect of the perturbation is only taken into account in the model operator (e.g. for the differential operations), and is neglected everywhere else. It implies that ensemble statistics (mean, standard deviation, covariance matrix,...) can be computed as usual on the reference grid, while the perturbed metrics must be used to compute any diagnostics involving a differential operator. In the following, for instance, the perturbed metrics were used to compute relative vorticity from the velocity fields, to be consistent with the perturbed model dynamics, which is specific to each member. For that purpose, the perturbed metrics were archived with time, at the hourly frequency, in each ensemble member.

### 3.2.1 Wave-number power spectrum

The stochastic scheme used in this work is designed to introduce uncertainty at the very small scales (on the order of 10 grid points, i.e. 10 km), and this uncertainty is then expected to develop and cascade toward larger scales spontaneously and consistently with the model dynamics. Figure 3 confirms that the stochastic perturbations do not alter the spectral characteristics of the physical quantities in the model. It compares the wavenumber power spectrum (Power Spectral Density, PSD) of SSH hourly snapshots in the different experiments, with or without stochastic perturbations, over a squared box of $L \sim 450$ km (box (a) in Fig.1). In average, over the 2 months of the experiments, the figure shows very consistent SSH spectra from the perturbed and unperturbed models.


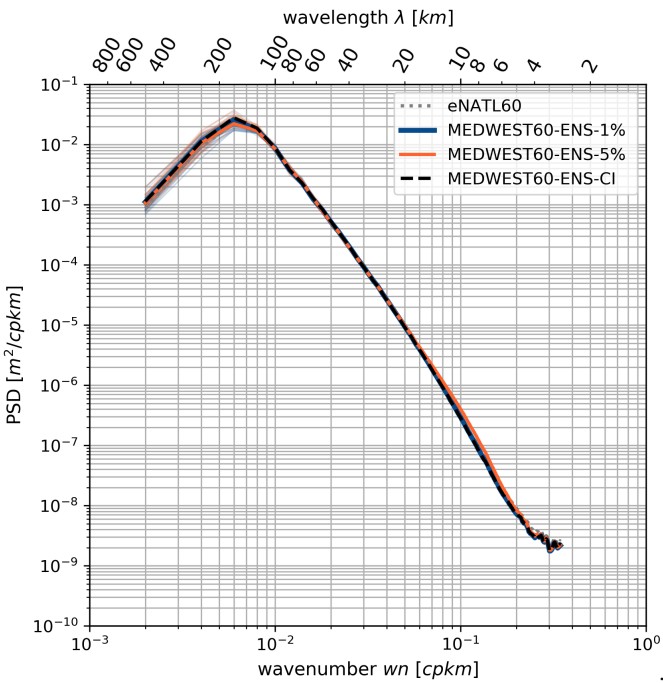

**Figure 3.** Wavenumber spectrum (Power Spectral Density, PSD) of hourly SSH snapshots in the MEDWEST60 ensemble experiments, in a box of $350 \times 350$ gridpoints, corresponding to $\sim 450 \times 450$ km (box (a) in Fig. 1). Comparison is also made with the eNATL60 simulation. The PSD of SSH $[m^2/cpkm]$ is averaged in time over 241 hourly snapshots of SSH, one hourly spectrum every 6h, over the 2 months of simulation and over all members of the given ensemble (thick lines). The PSD of all individual members are also shown in thin lines, in the same color as their ensemble mean.

Note that the spread of the PSD around the ensemble mean of each experiment is also shown in very thin lines in Figure 3:
the members all have a PSD very consistent with their ensemble mean (the spread is smaller than the thickness of the ensemble mean line) on all scales up to $\sim$150 km. For larger scales, some spread is seen between the members and it provides an idea of the sensitivity (significance) of such a spectral analysis on the last few point of the spectrum (aliasing effects). The spectra are computed here over a squared box of $L \sim$450 km (box (a) in Fig.1), and does not resolve well the spectral scales larger than $L/2$. The ensemble spread interval appearing in the figure thus provides some guidance as to interpret the significance of the
PSD variations in this scale range, and over this time period (a 2-month average here).

### 3.2.2   Growth of the ensemble spread

Figure 4 illustrates the evolution with time of the ensemble spread in the three ensemble experiments performed. The spread is computed here as the ensemble standard deviation of the hourly SSH, then spatially averaged over the entire MEDWEST60 domain, for each of the ensemble experiments. As expected, the ensemble spread initially grows faster in the perturbed exper-
iment with the large model error (ENS-5%) than with the smaller model error (ENS-1%) and in the unperturbed experiment





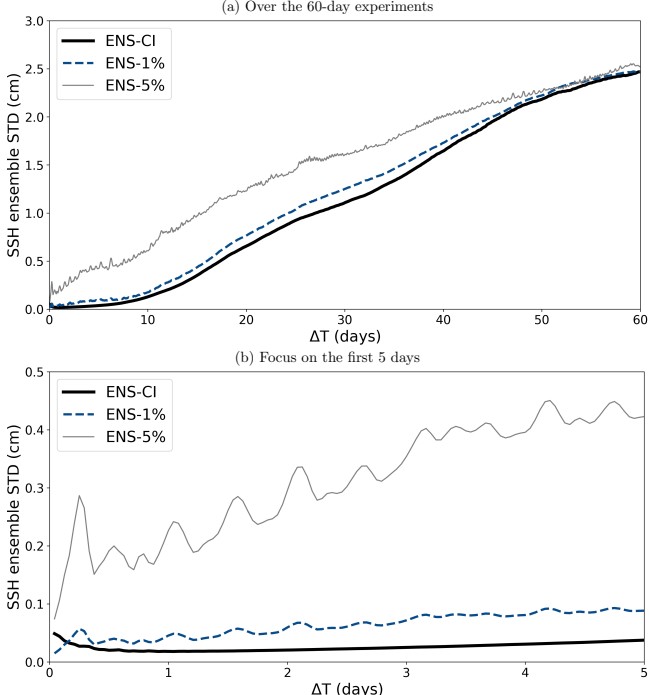

**Figure 4.** Time-evolution of the ensemble standard deviation of the hourly SSH, then spatially-averaged over the entire MEDWEST60 domain for the three ensemble experiments (ENS-1%, ENS-5%, ENS-1%-S, ENS-CI): (a) over 60 days, (b) focus on the first 5 days of simulation.

(ENS-CI). But after about 50 days of simulation, the ensemble spread of all three experiments (ENS-CI, ENS-1% and ENS-5%) have converged to a similar value. The spread is still growing at the end of the 60-day experiments but the curves have started to flatten, suggesting that our experimental protocol was successful at initiating divergent-enough ensembles on the targeted time-range (2 months). The saturation of the spread was further verified by extending one of the experiments by 2

more months (not shown here). Note also that similar characteristics of the spread growth have been seen in the other surface variables (SST, SSS, relative vorticity, not shown here).

After 2 months, the three experiments have reached an ensemble spread in SSH of about 2.5 cm in average over the domain, but local maxima of spread values are found around 10 cm (not shown). Those values are close to typical deviation values of hourly SSH over time in the Mediterranean region. Further investigations discussed in the following subsection (cf spatial

decorrelation), also confirm that the spatial decorrelation of the submeso- and meso- scale features has been reached by the end of the 2-month experiments.

After the first ∼10 days of simulation, the ensemble spread in the three ensembles evolves in a similar manner, at more or less the same rate, and almost linearly, until day 40-50, where the curves then start to flatten and converge. Only in the first few days, the presence of model uncertainty makes a difference in the growth rate, ENS-5% clearly showing a faster growth





than ENS-1% (the latter being slightly faster than ENS-CI in the very first few days). This result suggests that in the context of short-range forecasting (1-5 days), model uncertainties might play a role as much as uncertain initial conditions, and should be taken into account in operational systems.

Note that these experiments are initiated in the winter time (February) when mesoscale activity is expected to be the largest in the region. We have also performed an additional test experiment (not shown), identical to ENS-1% except for the start date, 220 taken in August, when mesoscale activity is expected to be low. We found that the growth of the ensemble spread in this case is significantly slower than with winter initial conditions. It is consistent with the idea that the seasonal level of mesoscale turbulence plays a significant role in the in ensemble spread of the forecast and thus in the quantification of predictability. In this paper we do not investigate the dependence to seasonality, as our main objective is to propose a methodology to quantify predictability as a function of both initial uncertainty and model uncertainty. We thus choose to focus on the winter season 225 where the dynamics of the system is expected to maximize the growth of the ensemble spread for a given initial uncertainty.

### 3.2.3 Spatial decorrelation

Figure 5 illustrates how the relative vorticity fields diverge with time in hourly snapshots from two different members of experiment ENS-CI. The focus is made here on a 250x250 grid-point subregion in order to better emphasize the smallest simulated ocean features. At short time-lag (+1 day), the ocean states in the two example members are barely distinguishable 230 from each other. With a +20 day time-lag, differences start to appear on the exact location of the small features and their shape. After 60 days, the differences have become more obvious even on larger features and eddies, and many features do not even have their corresponding feature in the other member. At the end of the experiment, the ocean state of the two members appear clearly distinct from each other.

In order to investigate more systematically the evolution with time of the spread between the members of an ensemble, and 235 its wavenumber spectral characteristics, we now consider the forecast "error", which we assess here as the difference in SSH hourly snapshots taken between all pairs of members in the ensemble, and at each time-lag. In other words, each member is alternatively taken as the truth, and compared to the 19 remaining members, taken as the ensemble forecast for that given truth. We then compute the power spectral density (PSD) of each pair difference at each time-lag, and then average over the 20 × 19 permuted pairs to obtain the systematic error. Figure 6 presents this mean PSD, charachterizing the systematic forecast 240 error, as a function of forecast time-lag, in all three ensemble experiments. For reference, on the same figure is also plotted the ensemble-mean PSD of the full-field SSH at time-lag +60 days.

After just one hour of simulation starting from perfect initial conditions with the probabilistic model in ENS-1% (yellow curve in Fig.6.b), the wavenumber spectrum of the forecast error peaks in the small scales around $\lambda$= 15 km and is still two orders of magnitude smaller than the level of spectral power in the full-field SSH (shown as reference in thick black line on the 245 figure). The same behaviour is observed for ENS-5% (Fig.6.c), except that the level of spectral power is one order of magnitude larger than in ENS-1% (since the spread grows faster in ENS-5% than in ENS-1%). With increasing time-lag, the shape of the PSD becomes more "red", with more spectral power cascading to the larger scales. By the end of all three experiments (+60-day time-lag), the PSD of the forecast error has almost converged to the reference full-field SSH PSD, suggesting that the





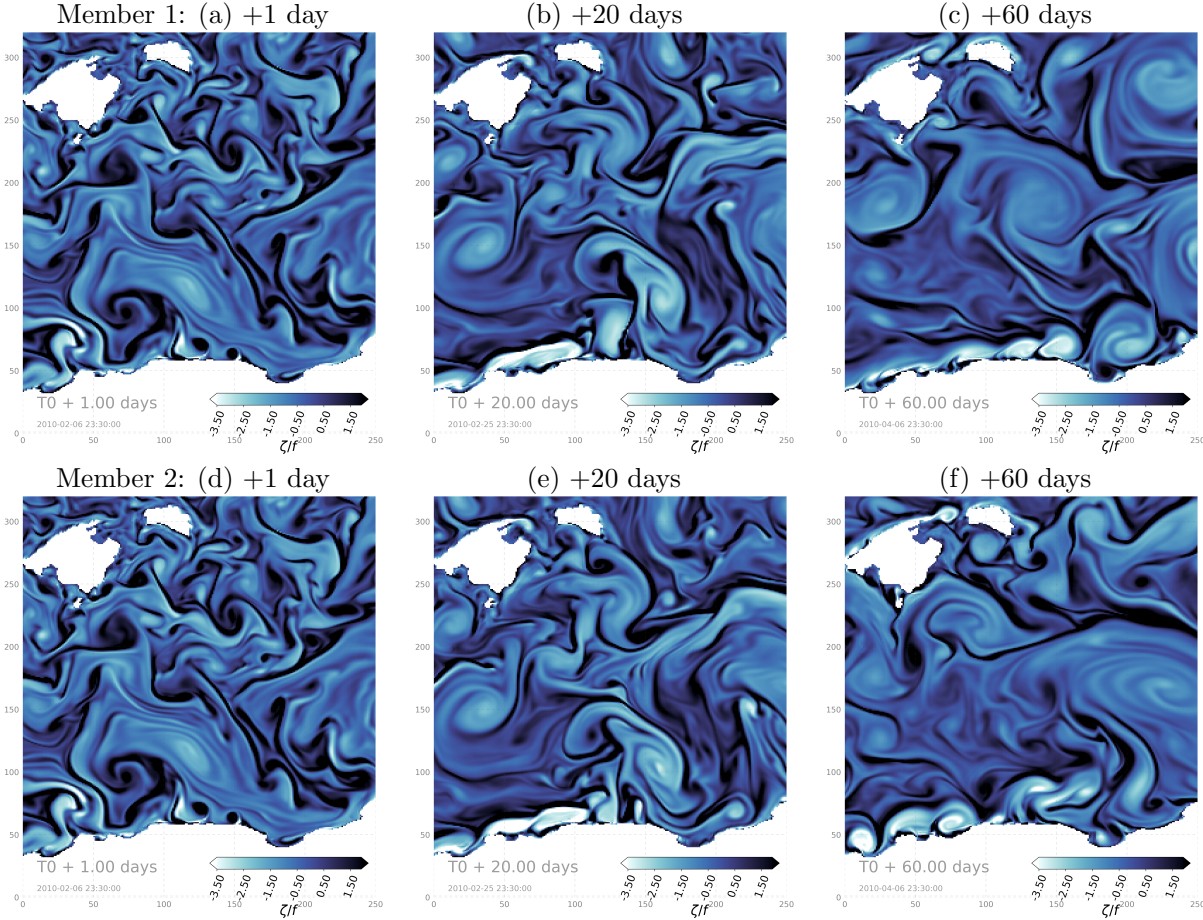

**Figure 5.** Hourly snapshots of relative vorticity from two example members (top and bottom, resp.) after 1, 20, 60 days (from left to right, resp.) in the ensemble experiment ENS-CI, focusing on a 250x250 gridpoint subregion south-east of the Balearic Islands (box (b) in Fig. 1).

members of each ensemble have become decorrelated on the spatial scale range considered here, i.e. 10-200 km. Note that we

do not necessarily expect a full spatial decorrelation between the members in this type of experiment since all members see

the same surface forcing and lateral boundary conditions. From Figure 6 it is also noteworthy that the evolution in time of the

forecast-error spectrum in ENS-CI (Fig. 6.a) and ENS-1% (Fig. 6.b) is very similar in amplitude and shape, except for the first

time-lag (+1 hour), where the curve in ENS-CI is already smoother than in ENS-1% and does not show the $\lambda$= 15 km peak as

in the latter. This is because ENS-CI is by design started from initial conditions from day +1 of ENS-1% (see section 3). In any

case, by a time-lag of 5 days, both ENS-CI and ENS-1% have converged to a very similar forecast-error spectrum and evolve

in the same manner.

   Overall, we thus find that after two months, the ensemble variance saturates over most of the spectrum, and the small scales

(<100 km) have become fully decorrelated between the ensemble members. This set of ensemble simulations is thus confirmed





to be appropriate to provide a statistical description of the dependence between initial accuracy and forecast accuracy for
time-lags between 1 and 20 days, consistently with the diagnostics proposed in the following.

## 4   Predictability diagnostics

In this section, we present predictability diagnostics where we quantify predictability, based on a given forecast score measuring
both the initial and forecast accuracy. Although any specific score of practical significance could have been used, we focus
here on a few simple and generic scores characterizing the *misfit* between ensemble members, in terms of (1) overall accuracy
(section 4.1: CRPS score), in terms of (2) geographical position of the ocean structures (section 4.2: location score) and in
terms of (3) spatial decorrelation of the small-scale structures (section 4.3: Decorrelation score).

### 4.1   Probabilistic score

A standard approach to evaluate the skill of an ensemble forecast using reference data (Candille and Talagrand, 2005; Candille
et al., 2007) is to compute probabilistic scores characterizing the statistical consistency with the reference (reliability of the
ensemble) and the amount of reliable information it provides (resolution of the ensemble). For instance, in meteorology,
ensemble forecasts can be evaluated a posteriori using the analysis as a reference. In the framework proposed in this study, a
consistent approach to assess predictability is thus to compute the probabilistic scores that can be expected for given initial and
model uncertainty. In this case, we can use one of the ensemble members as a reference, by assuming that it corresponds to the
true evolution of the system, and then compute the score using the remaining ensemble members as the ensemble forecast to
be tested. Furthermore, by repeating the same computation with each ensemble member as a reference, as in a cross-validation
algorithm, we can obtain a sample of the probability distribution for the score. All members of the ensemble are thus used
successively as a possible truth, for which the other members provide an ensemble forecast. This procedure is very similar to
the ensemble approach introduced in Germineaud et al. (2019) to evaluate the relative benefit of observation scenarios in a
biogeochemical analysis system. In this framework, the probabilistic score can be viewed as a measure of the resulting skill of
a given observation scenario.

### 4.1.1   CRPS score

A common measure of the misfit between two probability distributions of a one-dimensional random variable $x$ is the area
between their respective Cumulative Distribution Functions (CDF) $F(x)$ and $F_{\mathrm{ref}}(x)$:

$$\Delta = \int_{-\infty}^{\infty} \left| F(x) - F_{\mathrm{ref}}(x) \right| \, dx \tag{1}$$

In our application, the reference CDF $F_{\mathrm{ref}}(x)$ is a Heaviside function increasing by 1 at the true value of the variable, and
the ensemble CDF $F(x)$ is a stepwise function increasing by $1/m$ at each of the ensemble values (where $m$ is the size of the
ensemble). Thus the further the ensemble values from the reference, the larger $\Delta$, and the unit of $\Delta$ is the same as the unit of $x$.

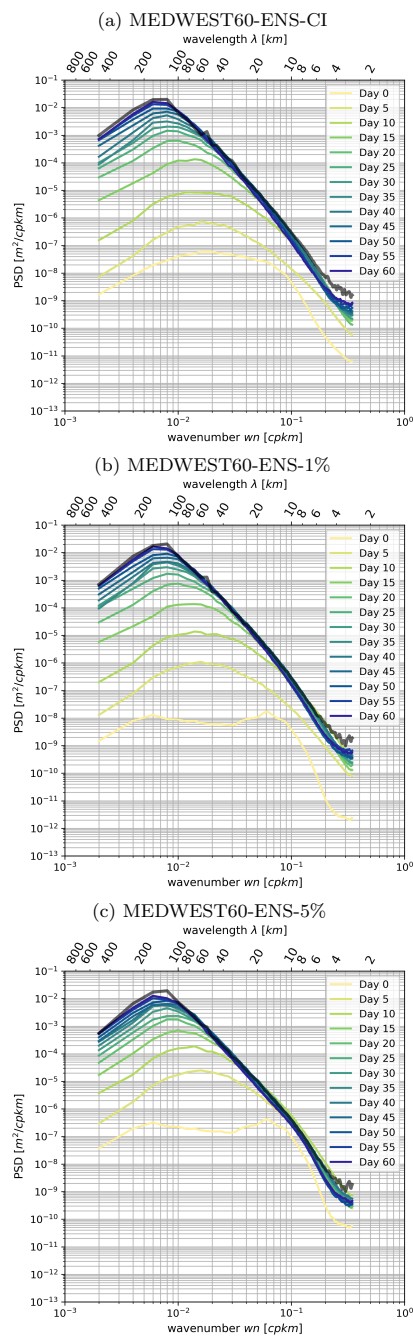

**Figure 6.** Ensemble-mean wavenumber power spectrum density (PSD) of the hourly SSH at day 60 (black thick line), compared to the mean PSD of the forecast error in experiments (a) ENS-CI, (b) ENS-1%, (c) ENS-5%. The forecast error is assessed as the difference of the hourly SSH fields between all pairs of members in the given ensemble, and the mean is taken of the PSDs of all the 20x19 permuted pairs at each time (time increasing from yellow to blue colors), see text in section 4.3 for more details. The time-lag labeled "Day 0" is taken after 1 hour of the experiment.





The continuous rank probability score (CRPS) is then defined (Hersbach, 2000; Candille et al., 2015) as the expected value of $\Delta$ over a set of possibilities. In practical applications, the expected value is usually replaced by an average of $\Delta$ in space

and time. In our application, the cross-validation algorithm would give the opportunity to make an ensemble average and thus be closer to the theoretical definition of CRPS. However, the ensemble size is here too small to provide an accurate local value of CRPS, so that we prefer computing a spatial average as would be done in a real system, and compute an ensemble of spatially-averaged CRPS scores. In the following, CRPS scores will be computed by averaging over a subregion of the domain basin south-east of the Balearic Islands (100x100 gridpoint region labelled as (c) in Fig. 1).

### 4.1.2 Evolution in time

We first investigate the ensemble experiment peformed with the deterministic model and uncertain (i.e. perturbed) initial conditions (ENS-CI). The additional effect of model uncertainties will be diagnosed in a second step.

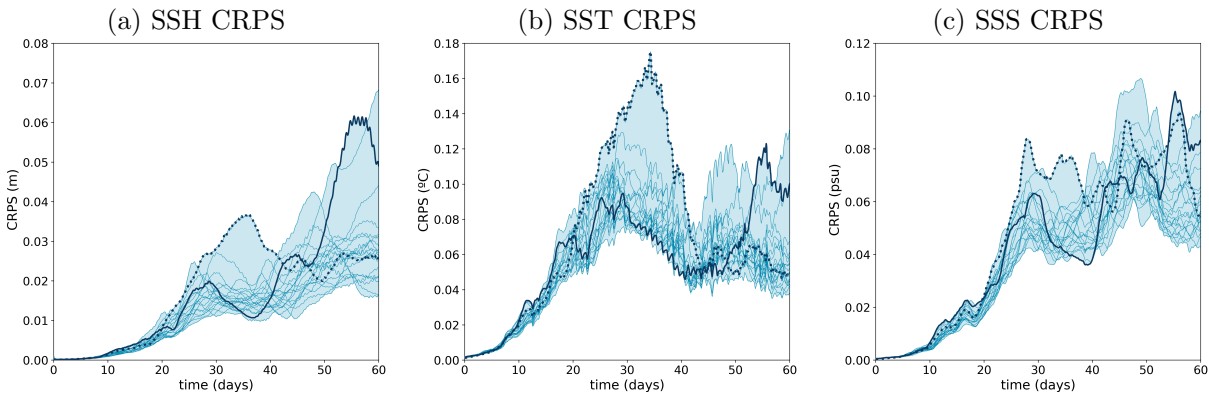

**Figure 7.** Time evolution of the CRPS score (y-axis) for SSH (meters), SST (degree Celsius) and SSS (psu) from experiment ENS-CI. The score is computed for each of the 20 permuted cases (thin lines) taking an ensemble member as the reference truth and the rest of the members as the ensemble forecast. The blue shading represents the min-to-max envelop of the 20 scores computed for the experiments. Two example lines are plotted thicker (solid and dotted lines) in each panel to illustrate how individual scores can evolve with time.

Figure 7 shows the time evolution of the CRPS score for SSH, SST and SSS as obtained in experiment ENS-CI. It is computed for each of the 20 permuted cases taking an ensemble member as the reference truth and the rest of the members as

the ensemble forecast. The CRPS score starts from zero and the initial increase is about exponential, with a doubling time of about 4 days. After typically 20 days, the evolution of the score becomes more irregular, globally increasing, but also sometimes decreasing in time, depending on the particular situation of the system. During the initial exponential increase, the diversity of possible evolutions of the score remains moderate: the score only increases a bit faster or a bit slower according to the member that is used as a reference. In the second period, however, the evolution becomes very diverse, with the score sometimes

increasing with time for a given reference member and decreasing for another reference member. This shows the importance of accounting for the diversity of possible situations in the description of predictability. With time, anomalous situations





can emerge, which can produce different predictability patterns. Predictability thus needs to be described as a probability distribution of the score for given conditions of initial uncertainty (and/or of model uncertainty).

### 4.1.3 Predictability diagrams

Using the time evolution of the ensemble CRPS score obtained in the previous section, it is then possible to describe predictability for a given time lag $\Delta t$ by the joint distribution of the initial and final score $CRPS(t)$ and $CRPS(t+\Delta t)$, respectively. From this distribution, we can indeed obtain the conditional distribution of the final score given the initial score, and reciprocally the conditional distribution of the initial score required to obtain a given final score.

Figure 8 describes predictability for 3 time lags $\Delta t = 2$, 5, and 10 days, for SSH, SST and SSS, by plotting the forecast
CRPS score (y-axis) conditioned on the initial CRPS score (x-axis) of the same variable. The figure is plotted for experiment ENS-CI, i.e.without model uncertainty. It is in fact just a reshuffling of the data from Figure 7, gathering all couples of scores with time lag $\Delta t$. It must be kept in mind that the figure mixes forecasts starting at a different initial time (in the range of the 2-month experiment), which can correspond to various situations of the system, in particular to different atmospheric forcings. The resulting probability distribution thus encompasses this set of possibilities, the only conditions being on the time lag $\Delta t$
and the initial CRPS score. To put a condition on the initial time would have required performing a large number of ensemble forecasts from that initial time with various levels of initial error, and would have been far too expensive.

The first thing to note from Figure 8 is that for a given initial score, there can be a large variety of final scores after a $\Delta t$ forecast, which again shows the importance of a probabilistic approach. What we obtain is a description of the probability distribution of the final score given the initial score, or reciprocally, the probability distribution of the initial score to obtain
a required final accuracy. These are just two different cuts (along the y-axis or along the x-axis) in the two-dimensional probability distribution displayed in the figure. From this probability distribution, it is then possible to compute the initial score required to have a 95% probability that the final score is below a given value. This result, corresponding to the green curve in the figure, can be viewed as one possible answer to the question raised in the introduction about the initial accuracy required to obtain a given forecast accuracy.

### 4.1.4 Effect of model uncertainties


To explore the potential additional effect of model uncertainties (as represented by the stochastic scheme described in section 2.2) on predictability, we can compare the CRPS diagnostics described above for the three ensemble experiments performed: ENS-CI (no model uncertainty), ENS-1% (moderate model uncertainty), and ENS-5% (larger model uncertainty). In that purpose, Figure 9 shows the time evolution of the CRPS score for these three experiments. We observe that forecast
uncertainty increase faster with model uncertainty included in the system (especially in ENS-5%), although the asymptotic behaviour of the score is very similar in all three simulations. Model uncertainty mainly matters for a short-range forecast (less than ~15 days) when the initial condition is very accurate. Of course, this conclusion only holds for the kind of location uncertainty that we have introduced in NEMO here, with short-range time and space correlation. A long standing effect of model uncertainty on predictability would be expected for large-scale perturbations, as in the atmospheric forcing, for example

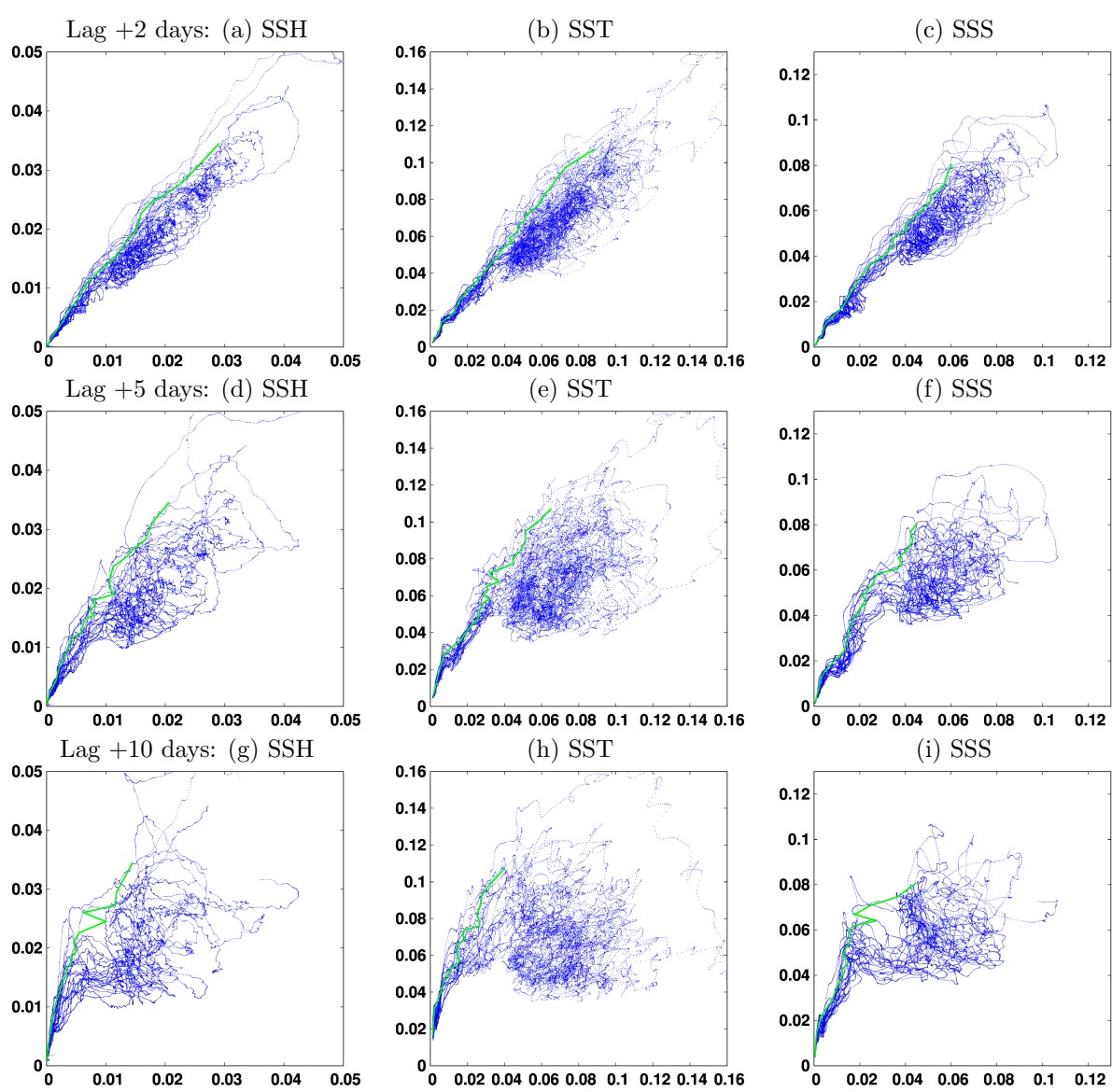

**Figure 8.** Final CRPS score (y-axis) as a function of the initial CRPS score (x-axis), for 3 time lags $\Delta t = 2$, 5, and 10 days (from top to bottom), for SSH (meters), SST (degree Celsius) and SSS (psu). The green line corresponds to the initial score required to have a 95% probability that the final score is below a given value.





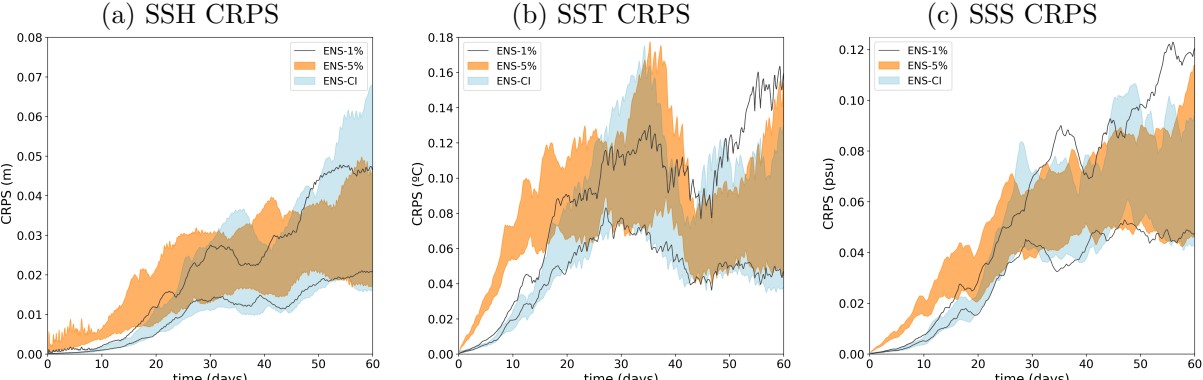

**Figure 9.** CRPS score as a function of time for (a) SSH (meters), (b) SST (degree Celsius) and (c) SSS (psu), compared for the three simulations: ENS-CI (no model uncertainty, in blue shading), ENS-5% (larger model uncertainty, in orange shading), and ENS-1% (small model uncertainty, in grey lines). Only the min-to-max envelop of all the 20 individual CRPS scores is represented for each experiment (the 20 individual scores are omitted).

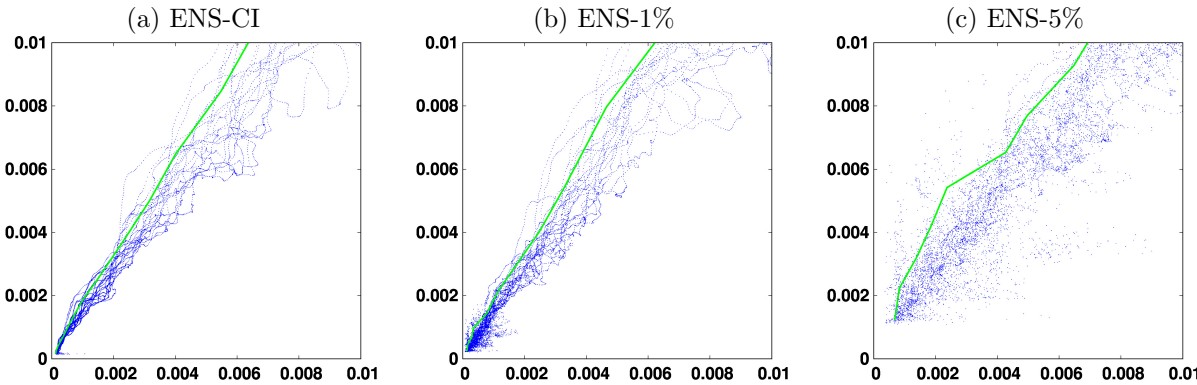

**Figure 10.** Forecast CRPS score (y-axis) as a function of the initial CRPS score (x-axis) for SSH (meters) at time lag $\Delta t = 2$ days. The green line corresponds to the initial score required to have a 95% probability that the final score is below a given value. The figure compares the three simulations (a) ENS-CI (no model uncertainty), (b) ENS-1% (small model uncertainty), and (c) ENS-5% (larger model uncertainty) for the small CRPS scores (smaller than 0.01 m).

(in this study though, we do not consider atmospheric forcing uncertainty, as the ocean model is forced by a prescribed and presumably "true" atmosphere).

The consequence of this specific impact of model uncertainty is that the predictability diagrams displayed in Figure 8 remain very similar for all three experiments, only becoming a bit more fussy when model uncertainty is included. To see the difference, we need to focus on the short time lag ($\Delta t = 2$ days) and on the small initial and final scores (which correspond to

the beginning of the experiments). Figure 10 compares the results obtained for SSH in ENS-CI, ENS-1% and ENS-5%, and we can observe that with larger model uncertainty, a smaller initial score (i.e. a more accurate initialization from observations)





is generally needed to obtain a given final score (i.e. a given target of the forecasting system). If this model uncertainty is irreducible (as argued in section 2.2 if it represents the effect of unresolved scales), they can thus represent an intrinsic limitation to predictability (at that resolution), at least in the specific case of a short time lag and a small initial error.

## 4.2 Location score

In the previous section, a probabilistic score has been used to describe the accuracy of the initial condition that can be associated to any given CMEMS observation/assimilation system. However, in many applications, what matters is not so much the accuracy of the value of the ocean variables, but the location of the ocean structures (fronts, eddies, filaments,...). Moreover, the acuteness of the positioning of ocean structures that can be obtained in the initial condition of the forecast can be thought to be rather directly related to the resolution of the observation system that is available for the operational forecast (in situ network or satellite imagery).

For these reasons, in this section, we will introduce a simple measure of location uncertainties in an ensemble forecast, which will be used in the same way as the CRPS score in the previous section. The same type of diagnostics will be computed to provide a similar description of predictability, but from a different perspective.

### 4.2.1 Misfit in field locations

To obtain a simple quantification of the position misfit between two ocean fields (one ensemble member and a reference truth), we are looking for an algorithm to compute at what distance the true value of the field can be found. Ideally, what we would like is to find the minimum displacement that would be needed to transform a given ensemble member into the reference truth. However, it is important to remark that this does not amount to computing the distance between corresponding structures in the two fields. This would indeed require an automatic tool to identify coherent structures in the two fields and would be much more difficult to achieve in practice. In general, if the two fields are not close enough to each other, such identification would even be impossible, since ocean structures can merge, appear, disappear or be transformed to such extent that no one-to-one correspondence can be found.

In addition, to further simplify the problem, we do not consider the original continuous fields, but modified fields that have been quantized on a finite set of values. Figure 11 shows for instance the salinity field from two example members of the ENS-CI simulation (after 15 days), together with their quantized version. The quantized version is obtained by computing the quantiles of the reference truth, for instance 19 quantiles here (from the distribution of all values in the map), and then by replacing the value of the continuous field by the index of the quantile interval to which it belongs (between 1 and 20). In this case, a value of 1 means that the field is below the 5% quantile and a value of 20 means that the field is above the 95% quantile. From these quantized fields, it is then easy to find the closest point where the index is equal to that of the reference truth, and thus where the field itself is close to the truth (to a degree that can be tuned by changing the number of quantiles).

Figure 12 shows the resulting maps of location misfit, in kilometers, for salinity in ENS-CI after 5, 10 and 15 days. We see that the location misfit increases with time as the two ensemble members diverge from each other. At +10 days, misfit values of 5-10 km are sparsely seen in the SSS field, featuring a thin elongated pattern that likely illustrates the arising misfit

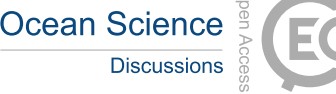

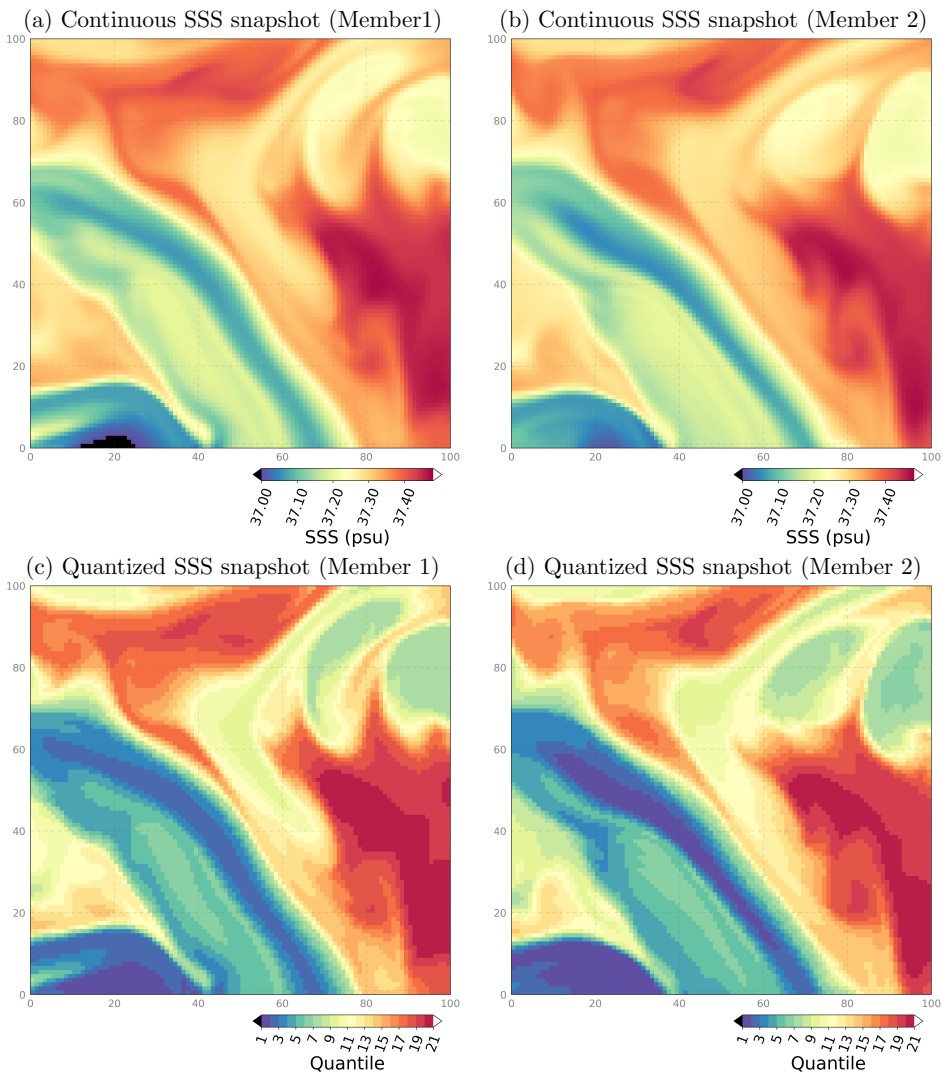

**Figure 11.** Surface sea salinity (SSS) hourly snapshots from two example members of experiment ENS-CI after 15 days, focusing on a 100x100 subregion south-east of the Balearic Islands: zoom (c) in Fig. 1. The original continuous fields are shown in the top row (a,b) and their quantized version (c,d) in the bottom row.

in the location of a NW-SE front (sharp gradient in the SSS field, as seen in Fig. 11.c-d). At +15 days, the location misfit has increased in amplitude and now covers most of the subregion, with maximum values of 15-20 kilometers. From such maps, it is then possible to define a single score from the distribution of distances. For the purpose of this study, we simply define our location score as the 95% quantile of this distribution, which means that location error has a 95% probability to be below the distance given by the score.

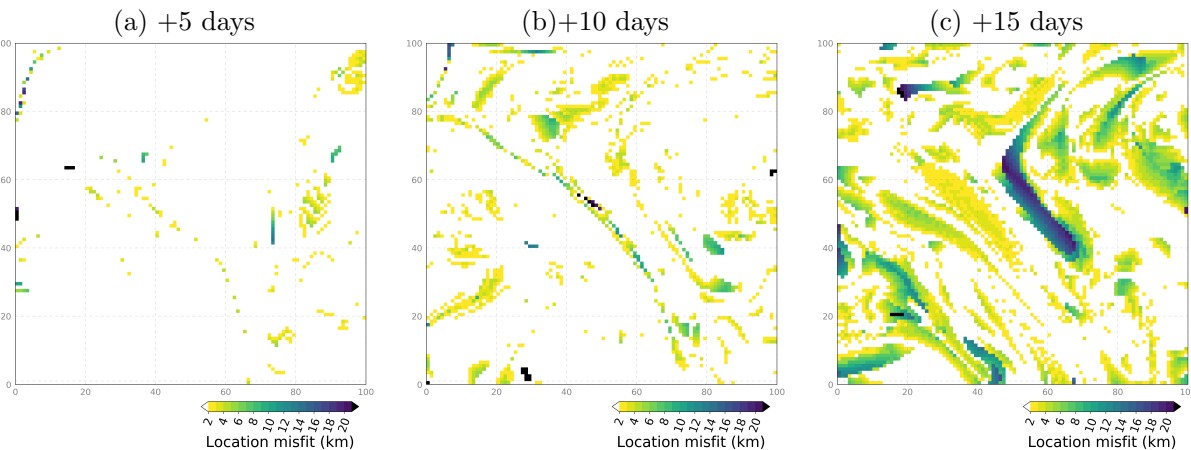

**Figure 12.** Location misfit (in km) between the surface salinity fields of two example members in the ensemble experiment ENS-CI after 5, 10 and 15 days (from left to right), focusing on a subregion south-east of the Balearic Islands: zoom (c) in Fig. 1.

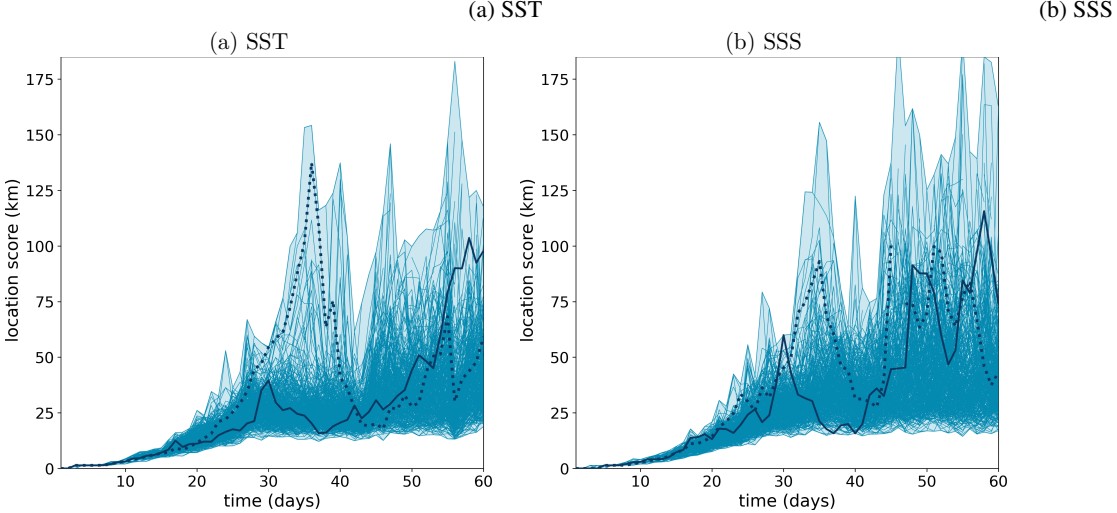

**Figure 13.** Time evolution of the location score (y-axis, in km) for (a) SST and (b) SSS in experiment ENS-CI. The score is computed for each of the 20x19 permutted pairs (blue lines) considering one ensemble member as the reference truth and each of the 19 remaining ensemble members as a forecast. The blue shading represents the min-to-max envelop of all the 20x19 scores computed for the experiment. Two example lines are plotted thicker (solid and dotted lines) in each panel to illustrate how individual scores can evolve with time.

**4.2.2 Evolution in time**

We start by analyzing the time evolution of the location score in ensemble experiment ENS-CI, where the only source of uncertainty comes from the initial conditions (no model uncertainty; the model is deterministic). Figure 13 shows the evolution in time of the location score for SST and SSS, considering each pair of members in the ensemble, which amounts to a total

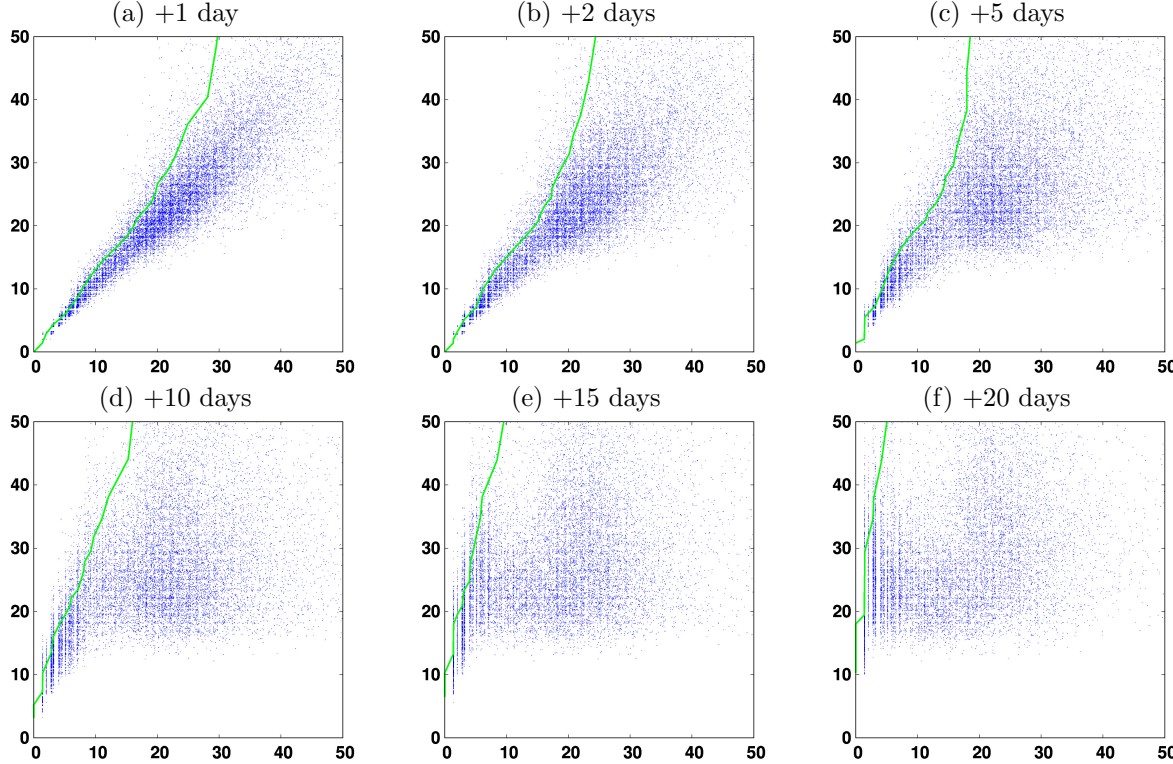

**Figure 14.** Final SST location score (y-axis, in km) as a function of the initial SST location score (x-axis, in km) for experiment ENS-CI, for 6 time lags $\Delta t = 1, 2, 5, 10, 15$ and 20 days (from top left to bottom right). The green line corresponds to the initial score required to have a 95% probability that the final score is below a given value.

of $m(m-1) = 20 \times 19 = 380$ curves displayed in the figure. The two panels, for (a) SST and (b) SSS, both show a similar
distribution of the time evolutions, confirming that our quantification of location uncertainty is consistent for these two tracers.
Figure 13 also shows that during about the first half of the experiment (the first 30 days), the location score increases towards
saturation, with a spread that also increases with time, whereas in the second half of the experiment, the score has reached the
asymptotic distribution, which is characterized by a large location uncertainty and a large spread of the score. It means that
there is no more information about the location of the ocean structures in the forecast and that the score can be either moderate
(down to 20 km) or very large (up to 80 km and more) depending on chance. In the following, we thus mostly focus on the
range of scores, between 0 and 20 km, where a valuable forecast skill can be expected (for the small-scale tracer structures that
are resolved by the model).

### 4.2.3 Predictability diagrams

From the time evolution of the score described in the previous section, we can then deduce predictability diagrams, following
exactly the same approach as for CRPS in section 4.1.3. Figure 14 describes predictability (computed from SST fields) for





6 time lags ($\Delta t = 1$, 2, 5, 10, 15 and 20 days), by showing the final location score (y-axis) as a function of the initial location score (x-axis). Again, this figure is just a reshuffling of the data from Figure 13, gathering all couples of scores with time lag $\Delta t$, using the same assumption already discussed in section 4.1.3. Note that the longest time-lags considered here (>10 days) are relevant only in the present context of forced ocean experiments (as a forecasted atmosphere would also become a

major source of uncertainty for ocean predictability in a real operational forecast context at those time lags).

The interpretation of Figure 14 follows the same logic as the previously discussed predictability diagrams for CRPS. But the structure of the diagrams is here even more directly understandable, and the loss of predictability with time appears more clearly. For instance, if one seeks a forecast accuracy of 10 km with a 95% confidence (i.e. a y-value of the green curve equal to 10 km), then Figure 14 tells that the initial location accuracy required (necessary condition, but not sufficient, see

the conclusion section) is about 8 km for a 1-day forecast, 6 km for a 2-day forecast, 4 km for a 5-day forecast, 2 km for a 10-day forecast, and that this target is impossible to achieve in a 15-day and 20-day forecast. In the two latter cases however, the impossibility to achieve the targeted accuracy might just be due to the absence of small-enough initial errors in our sample (since ENS-CI was initialized using ENS-1% after 1 day). But this should not make any practical difference since such small initial errors would anyway be impossible to obtain in a real system.

### 4.2.4   Effect of model uncertainties

As for the CRPS score, we then explore the possible additional effect of model uncertainty, by comparing the results from experiment ENS-CI (no model uncertainties) with those from ENS-1% (small model uncertainties) and ENS-5% (larger model uncertainties). Figure 15 compares first the time evolution of the location score for ENS-CI (in blue) and ENS-5% (in orange), and we observe again that model uncertainty mainly matters at the beginning of the simulation by a faster increase of the

forecast uncertainty, towards a similar asymptotic behaviour for the two simulations.

As for the CRPS score, the predictability diagrams are only substantially different between the experiments for short time lags and small initial and final scores. This is illustrated in Fig. 16 by comparing the predictability diagrams obtained for SST in (a) ENS-CI, (b) ENS-1% and (c) ENS-5% for $\Delta t = 5$ days and scores below 20 km. Again, we can detect here a moderate effect of model uncertainty (as simulated here) on predictability. For instance, if one seeks a forecast accuracy of 10 km with a

95% confidence, the initial location accuracy required decreases from about 4 km in ENS-CI to about 3 km in ENS-5%.

As expected, our results show that the initial location accuracy plays a major role in driving the forecast location accuracy, but irreducible model uncertainties can also play a role for short time lags and accurate initial conditions.

### 4.3   Decorrelation score

To complement the information provided by the location score above on the "misfit" of the ocean structures, we also investigate

the decorrelation of the ensemble members in spectral space.



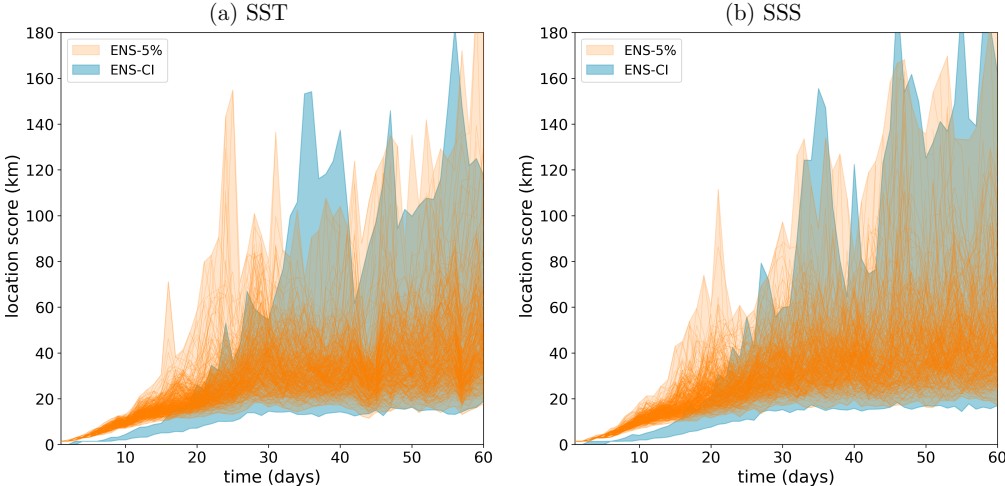

**Figure 15.** Time evolution of the location score (y-axis, in km) for (a) SST and (b) SSS, from two of the experiments: ENS-CI (no model uncertainty, in blue) and ENS-5% (larger model uncertainty, in orange). Only the min-to-max envelop of all the 20x19 individual location scores is represented for experiment ENS-CI as the individual 20x19 individual scores are already plotted in Fig. 13. The individual scores and the min-to-max envelopp are superposed here for experiment ENS-5%.

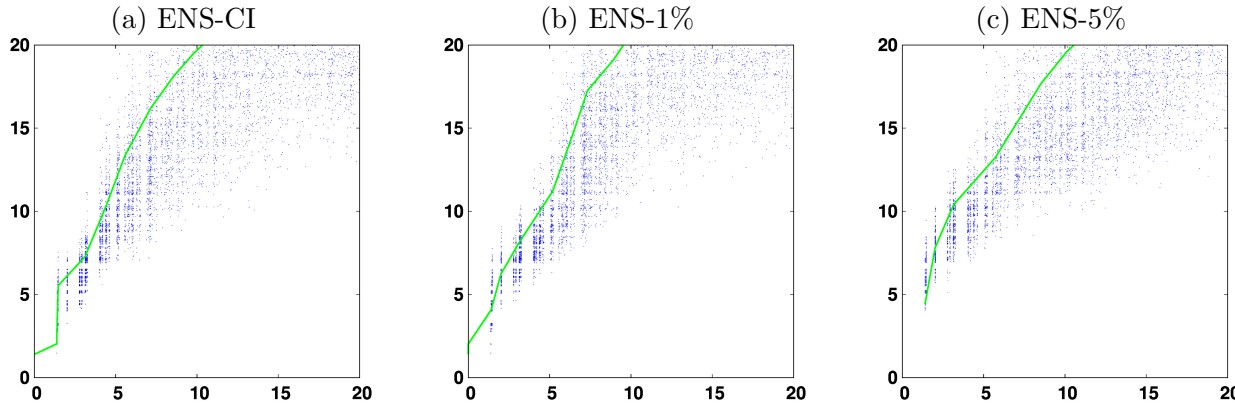

**Figure 16.** Final location score (y-axis, in km) as a function of the initial location score (x-axis, in km) for SST and time lag $\Delta t = 5$ days. The green line corresponds to the initial score required to have a 95% probability that the final score is below a given value. The figure compares the three simulations ENS-CI (no model uncertainty, left panel), ENS-1% (small model uncertainty, middle), and ENS-5% (larger model uncertainty, right panel) for the small location scores (smaller than 20 km).

### 4.3.1 Decorrelation as a function pf spatial scale

The idea behind this additional score is to compare the spectral content of the forecast error to the spectral content of the reference field (here considering SSH). The forecast error is assessed as the difference of SSH maps (hourly averaged) between





a given member taken as the truth and another member considered as the forecast. All the 20x19 combinations of pairs are
alternatively considered, following the same cross-validation algorithm as described in the previous sections. The "misfit" of
the ocean structures is here quantified in spectral space with a ratio R of decorrelation, computed for each time-lag as:

$$R = 1 - \frac{<\mathrm{PSD}_{\mathrm{diffssh}}>}{2\times <\mathrm{PSD}_{\mathrm{ssh}}>}, \tag{2}$$

where $\mathrm{PSD}_{\mathrm{ssh}}$ is the Power Spectral Density of the full-field SSH at that given time-lag, and $\mathrm{PSD}_{\mathrm{diffssh}}$ is the PSD of the
forecast error on SSH at that given-time-lag. The brackets <...> denote the ensemble mean operation over the 20 members or
over the 20×19 combinations of pairs. The PSDs are computed in the squared box of $L \sim$450 km shown as box (a) in Fig.1. By
design, R is expected to tend to zero when the ensemble members are fully decorrelated, and to be close to 1 when the members
are fully correlated. The factor 2 in the definition of R comes from the fact that we compare here the PSD of a difference of two
given fields with the PSD of the reference field. For example, if the ensemble members are strictly independent and uncorrelated
in space on all scales, then for all combinations of a pair of members $(t, f)$ where $t$ would be considered the truth and $f$ the
forecast, the space variance (var) of the difference $f - t$ can be expressed as :

$$<\mathrm{var}(f - t)> = <\mathrm{var}(f) + \mathrm{var}(t) - 2\mathrm{covar}(f,t)>, \tag{3}$$

$$<\mathrm{var}(f - t)> = <\mathrm{var}(f)> + <\mathrm{var}(t)>, \tag{4}$$

$$<\mathrm{var}(f - t)> = 2 <\mathrm{var}(f)>, \tag{5}$$

where the factor 2 appears.

### 4.3.2   Evolution in time

In section 3.2, we have already discussed the evolution with time of the spatial spectral content of the forecast error (Figure
6). Now Figure 17 shows the evolution in time of the ratio R, computed at different time-lags from experiment ENS-CI in
the top panel. By design, values of R are close to 1 when the members are strongly correlated: this is indeed the case on the
figure, at very short time lags (<5 days, yellow line). With time increasing, R decreases (the members are less and less spatially
correlated), starting from small scales and cascading to larger scales. At the end of the 2-month experiment, R has decreased
to zero for scales in the range 10-60 km, consistently with what we had already deduced from Figure 6. Full decorrelation is
not yet reached for larger scales, but we do not necessarily expect a full spatial decorrelation between the members in this type
of experiment since all members see the same surface forcing and lateral boundary conditions. Also, note that the size of the
box on which the spatial spectral analysis is performed is about 350 km square, so the left part of the spectrum is not expected
to be much significant for scales larger than $\sim$150 km (aliasing effect, also see Fig.3 and associated text).

On the right side of the spectrum, on very small scales (<6 km), it is noteworthy that R remains larger than 0.5 after 2 months
of simulation. This behavior is consistent in the three experiments (see panels a,b,c in Fig.17), so it cannot just result from a
spurious effect of the stochastic perturbation (which is not present in experiment ENS-CI). Specific investigations would be
needed to understand better the reasons for this behavior, but note that it might just result from numerical truncation errors,
given the small amplitude of the signal (see Fig. 6) on the range of scales considered here (<6 km).



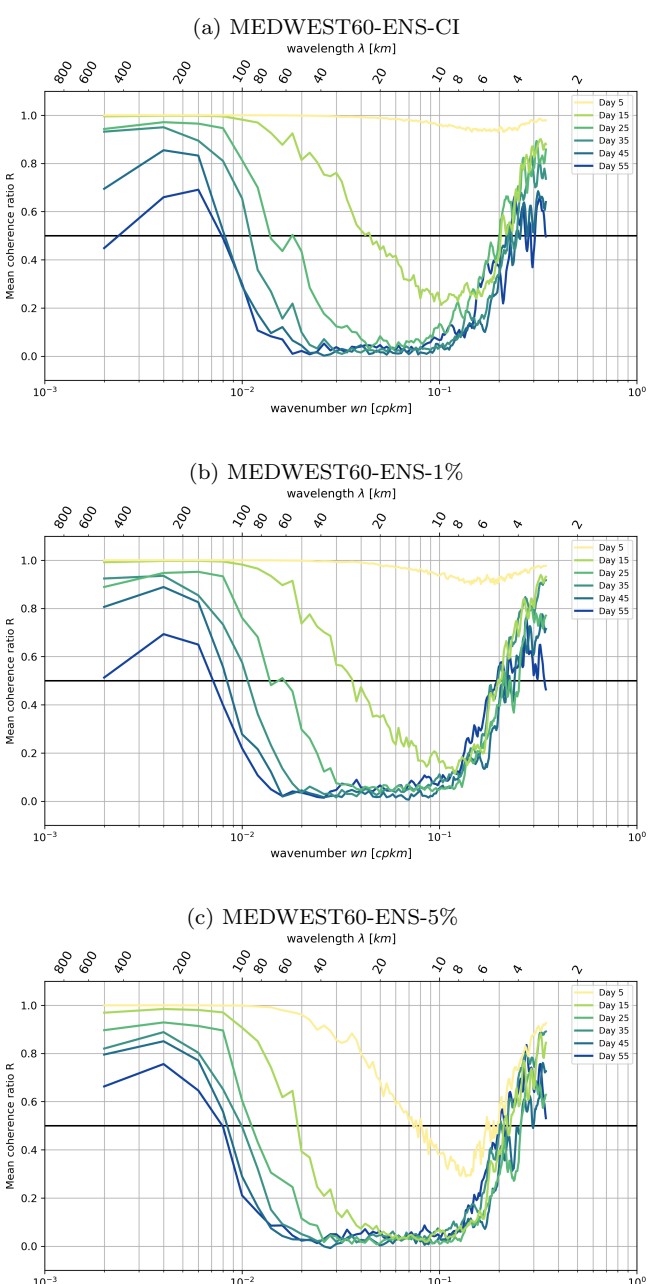

**Figure 17.** Mean coherence ratio R (see text for definition) from experiments ENS-CI, ENS-1% and ENS-5%. The ratio is computed at different time-lags: time increasing from yellow to blue colors.





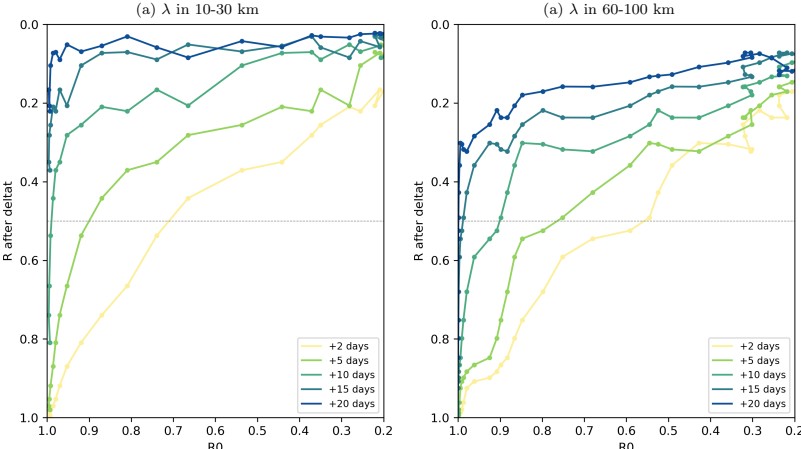

**Figure 18.** Mean wavenumber spectral coherence ratio R of the ensemble forecast as a function of the coherence of the ensemble initial conditions, for different forecast time-lags (+2,5,10,15,20 days), computed from hourly SSH in experiment ENS-CI. The mean ratio R is taken over scales of (a) 10-30 km and (b) 60-100 km in (b). A gray horizontal line marks the value of the coherence ratio R at 0.5, at which we consider the decorrelation of the ensemble members as effective.

### 4.3.3 Evolution in time and predictability diagrams

To provide an example of predictability diagram based on this spectral analysis, we finally consider the mean ratio $\overline{R}$, averaged over two given ranges of scales (10-30 km and 60-100 km) from experiment ENS-CI in Figure 18, following the same methodology as for the CRPS and location scores. The value of $\overline{R}$ after a given forecast time-lag, $\overline{R}(t+\Delta)$, where $\Delta$ is the time-lag,

is plotted as a function of the initial value $\overline{R}(t)$. The figure thus provides, for each given scale range ((a) 10-30 km and (b) 60-100 km), some objective information about the spatial decorrelation between the members (here in the case with no model uncertainty).

In the 10-30 km scale range for example, it appears that even with very small initial errors (initial R close to 1), the members become nearly decorrelated after a time-lag of ∼10 days (i.e. $\overline{R}(t+\Delta)$<0.5) on these scales. For the larger-scale range, 60-100

km, the threshold of $\overline{R}(t+\Delta)$<0.5 is reached for time lags above ∼15 days. Note however that only the uncertainty on initial conditions is taken into account here. A faster decorrelation would be expected if other types of uncertainties in the forecast system were taken into account, such as uncertainty on the atmospheric forcing.

The kind of predictability diagrams proposed in Figure 18 might also be relevant in the context of preparing for the assimilation of wide-swath high-resolution satellite altimetry such as expected from the future SWOT mission (Morrow, 2019). This

mission is expected to measure sea surface height (SSH) with high-precision and resolve short mesoscale structures as small as 15 km on a wide swath of 120 km. However the time interval between revisits will be within 11 to 22 days, depending on the location. Our results above tend to show that, for time-lags longer than 10 days, the forecasting system considered in the present study will have lost most of the information in the initial condition regarding SSH structures in the smallest scale range



(10-30 km). This is why nadir altimeter data will remain a key component of the satellite constellation complementing the
wide-swath SWOT measurements in space and time.

## 5 Summary and conclusions

The general objective of this study was to propose an approach to quantify how much of the information in the initial condition
a high-resolution NEMO modelling system is able to retain and propagate correctly during a short and medium range forecast.

For that purpose, a kilometric-scale, NEMO-based regional model for the Western Mediterranean (MEDWEST60, at 1/60°
horizontal resolution) has been developed. It has been defined as a subregion of a larger North Atlantic model (eNATL60),
which provides the boundary conditions at hourly frequency at the same resolution. This deterministic model has then been
transformed into a probabilistic model by introducing an innovative stochastic parameterization of location uncertainties in the
horizontal displacements of the fluid parcels. The purpose is primarily to generate ensemble of initial conditions to be used in
the predictability studies, and it has also been applied to assess the possible impact of irreducible model uncertainties on the
skill of the forecast.

With this regional model, 20-member and 2-month ensemble experiments have been performed, first with the stochastic
model for two levels of model uncertainty, and then with the deterministic model from perturbed initial conditions. In all
experiments, the spread of the ensemble emerges from the small scales (10 km wavelength) to progressively develop and
cascade upscale to the largest structures. After two months, the ensemble variance has saturated over most of the spectrum
(10-100 km) and the ensemble members have become decorrelated in this scale range. These ensemble simulations are thus
appropriate to provide a statistical description of the dependence between initial accuracy and forecast accuracy over the full
range of potentially useful forecast time lags (typically, between 1 and 20 days).

From these experiments, predictability has then been quantified statistically, using a cross-validation algorithm (i.e. using
alternatively each ensemble member as a reference truth and the remaining 19 members as forecast ensemble) together with
a few example scores to characterize the initial and forecast accuracy. From the joint distribution of initial and final scores, it
was then possible to diagnose the probability distribution of the forecast score given the initial score, or reciprocally to derive
conditions on the initial accuracy to obtain a target forecast skill. Although any specific score of practical significance could
have been used, we focused here on simple and generic scores describing the misfit between ensemble members in terms of
overall accuracy (CRPS score), geographical position of the ocean structures (location score), and spatial decorrelation.
Tables 3 and 4 give a quantitative illustration of the conditions obtained on the initial accuracy to obtain a given forecast
accuracy if the model is assumed perfect (as in experiment ENS-CI), using the CRPS score and the location score. For example,
Table 4 shows that, for our particular region and period of interest, the initial location accuracy required with a perfect model
(deterministic operator) to obtain a forecast location accuracy of 10 km with a 95% confidence is about 8 km for a 1-day
forecast, 6 km for a 2-day forecast, 4 km for a 5-day forecast, 1.5 km for a 10-day forecast, and that this target is unreachable
for a 15-day and a 20-day forecast (more precisely, in these two cases, the required initial accuracy would be irrealistically small





| Target forecast score (°C) | 2 days | 5 days | 10 days |
|---|---|---|---|
| 0.025°C | 0.016 | 0.006 | 0.001 |
| 0.05°C | 0.037 | 0.027 | 0.010 |
| 0.075°C | 0.056 | 0.039 | 0.023 |
| 0.1°C | 0.077 | 0.059 | 0.033 |

**Table 3.** Initial SST accuracy required (CRPS score, in °C) to obtain the target final accuracy (CRPS score, in °C, left column) with a 95% confidence for different forecast time lags: 2 days, 5 days and 10 days.

| Target forecast score | 1 day | 2 days | 5 days | 10 days | 15 days | 20 days |
|---|---|---|---|---|---|---|
| 2 km | 1.6 km | 1.4 km | — | — | — | — |
| 5 km | 3.9 km | 3.1 km | 1.4 km | — | — | — |
| 10 km | 7.9 km | 6.2 km | 4.4 km | 1.4 km | — | — |
| 15 km | 11.7 km | 10.4 km | 6.3 km | 3.1 km | 1.4 km | — |
| 20 km | 16.2 km | 14.9 km | 10.5 km | 5.4 km | 2.3 km | 1.4 km |

**Table 4.** Initial location accuracy required (location score, in km) to obtain the target final location accuracy (location score in km, left column) with a 95% confidence for different forecast time lags between 1 day and 20 days.

and was not included in our sample). With model uncertainties (stochastic operator, as in experiment ENS-1% or ENS-5%), the requirement on the initial condition can be even more stringent, especially for a short-range and high-accuracy forecast.

However, it is important to remember that this only provides *necessary* conditions but not a *sufficient* conditions on the initial model state. The reason for that is that the condition is put on one single score for one single variable, whereas the quality of the forecast obviously depends on the accuracy of all variables in the model state vector. In the examples given in the tables, we used the same model variable for both target score and the condition score, but we could have looked as well for a necessary condition on another variable (for instance velocity) to obtain a given forecast accuracy for SST or any other model diagnostic. In this way, for any forecast target, we could have accumulated many necessary conditions on various key properties of the initial conditions, especially observed properties, but this would never become a sufficient condition.

Furthermore, these necessary conditions on observed quantities can then be translated into conditions on the design of ocean observing systems, in terms of accuracy and resolution, if a given forecast accuracy is to be expected (e.g. on the SWOT altimetry mission, as discussed in section 4.3).

But, again, these conditions are only necessary conditions, as the accuracy of the initial model state also depends on the ability of the assimilation system to interpret properly the observed information and to produce an appropriate initial condition for the forecast. Checking this ability would have required performing observation system simulation experiments (OSSE) using the operational assimilation system, which lied beying the scope of the present work.



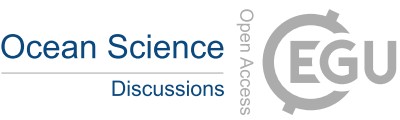

More generally, however, what this study suggests is that an ensemble forecasting framework should become an important component of CMEMS systems to provide a systematic statistical quantification of the relation between the system operational target (a useful forecast skill) and the available assets: the observation systems, with their expected resolution and accuracy, 535 and the modelling tools, with their target resolution and associated irreducible uncertainties.

*Code and data availability.* The source codes of the MEDWEST60 NEMO-based model and some of the diagnostics tools developed in this study are shared in an open github repository dedicated to MEDWEST60: https://github.com/ocean-next/MEDWEST60. The model outputs are available upon request (∼50 To for the three ensemble simulations). The CDFTOOLS (https://github.com/meom-group/CDFTOOLS) were used for some of the pre- and post-processing of the model outputs. The Power Spectral Density computation in sections 3.2 and 540 4.3 were all performed using A. Ajayi's python module PowerSpec (https://github.com/adeajayi-kunle/powerspec) and the predictability diagnostics based on CRPS score and location score were made with the SESAM (https://github.com/brankart/sesam) and EnsDAM (https://github.com/brankart/ensdam) softwares.

## Appendix A: Location uncertainties

The purpose of this appendix is to describe the stochastic parameterization that has been used in this paper to simulate model 545 uncertainties in experiments ENS-1% and ENS-5%. Uncertainties are assumed to occur on the location of the fluid parcels as explained in section A1. The further assumptions that are made to implement the resulting stochastic formulation in NEMO are presented in section A2.

### A1 Stochastic formulation

Location errors in a field $\varphi(\mathbf{x},t)$, function of the spatial coordinates $\mathbf{x}$ and time $t$, occur if the field $\varphi$ displays the correct values 550 but not at the right location. More precisely, this means that the field $\varphi(\mathbf{x},t)$ can be related to the true field $\varphi^t(\mathbf{x},t)$ by the transformation:

$$\varphi^t(\mathbf{x},t) = \varphi\left[\mathbf{x}^t(\mathbf{x},t),t\right] \tag{A1}$$

where $\mathbf{x}^t(\mathbf{x},t)$ is an anamorphic transformation of the coordinates defining the location where to find the true value of $\varphi(\mathbf{x},t)$. With respect to the true field $\varphi^t$, the values of $\varphi$ are thus shifted by:

$$\delta\mathbf{x}(\mathbf{x},t) = \mathbf{x}^t(\mathbf{x},t) - \mathbf{x} \tag{A2}$$

which defines the location error.





If the field $\varphi(\mathbf{x}, t)$ is evolved in time, over one time step $\Delta t$, with the model $\mathcal{M}$:

$$\varphi(\mathbf{x}, t + \Delta t) = \mathcal{M}[\varphi(\mathbf{x}, t), t] \tag{A3}$$

we can make the assumption that one of the effect of the model is to generate location uncertainties. In an advection-dominated
regime, this means for example that the displacement of the fluid parcels can be different from what the deterministic model
predicts. With this assumption, the model transforms to:

$$\varphi[\mathbf{x} + \delta\mathbf{x}(\mathbf{x}, t + \Delta t), t + \Delta t] = \mathcal{M}\{\varphi[\mathbf{x} + \delta\mathbf{x}(\mathbf{x}, t), t], t\} \tag{A4}$$

where the location error $\delta\mathbf{x}(\mathbf{x}, t)$ can be simulated for instance by a stochastic process $\mathcal{P}$:

$$\delta\mathbf{x}(\mathbf{x}, t + \Delta t) = \mathcal{P}[\delta\mathbf{x}(\mathbf{x}, t), \varphi(\mathbf{x}, t), t] \tag{A5}$$

where an explicit dependence to $\varphi$ and $t$ has here been included to keep the formulation general.

In ocean numerical models, the coordinates $\mathbf{x}$ are usually discretized on a constant grid. To implement the stochastic model of
Eq. (A4) and Eq. (A5) on this numerical grid, one possibility would be to remap the updated field $\varphi[\mathbf{x} + \delta\mathbf{x}(\mathbf{x}, t + \Delta t), t + \Delta t]$ on
this constant grid at each model time step. This remapping would correspond to a stochastic shift of the model field accounting
for the presence of location uncertainties. However, this solution may be computationally ineffective, and it is much easier to
keep track of the modified location of the grid points (described by $\delta\mathbf{x}$), and use this modified grid to implement the model
operator $\mathcal{M}$. In practice, to avoid deteriorating the model numerics, this solution requires that location errors remain small with
respect to the size of the grid cells, and that their variations over one time step $\Delta t$ are kept small enough to avoid undesirable
numerical effects.

This simple approach to simulate location uncertainties in ocean models has a close similarity to the work of Mémin (2014);
Chapron et al. (2018), where it is argued that the effect of unresolved processes in a turbulent flow can be simulated by adding
a random component to the Lagrangian displacement $\mathbf{dX}$ of the fluid parcels (as in a Brownian motion):

$$\mathbf{dX} = \mathbf{v}(\mathbf{x}, t) \, dt + \boldsymbol{\sigma}(\mathbf{x}, t) \, \mathbf{dB} \tag{A6}$$

where $\mathbf{v}(\mathbf{x}, t)$ is the velocity (as resolved by the model), $\mathbf{dB}$ is a stochastic process correlated in space but uncorrelated in time,
and $\boldsymbol{\sigma}(\mathbf{x}, t)$ defines the amplitude of the random displacement. The purpose of their study is then to examine the effect of
this modified material derivative (with the stochastic displacement added) when transformed into an Eulerian framework (i.e.
in a constant coordinate system). In a nutshell, from this assumption, the authors manage to derive modified Navier-Stokes
equations, with additional deterministic and stochastic terms depending on $\boldsymbol{\sigma}$.





## A2 Implementation in NEMO

To implement location uncertainties in NEMO, we explicitly make the assumption that the location errors $\delta \mathbf{x}$ remain small with
respect to the size of the grid cells, so that the nodes of the modified grid just follow a small random walk around the nodes of the original grid. Consistently with this assumption, we make the approximation that the model input data (bathymetry, atmospheric forcing, open-sea boundary conditions, river runoffs,...) keep the same location with respect to the model grid, which means that these data are not remapped on the moving grid. Such a tiny shift of the data (much smaller than the grid resolution) would indeed represent a substantial computational burden, with many possible technical complications, and would
only produce small additional perturbations to the model solution, which do not correspond to the main effect that we want to simulate.

Since the model grid is assumed fixed with respect to the outside world, we need only represent the displacement of each model grid point relative to its neighbours. In NEMO, this relative displacement of the model grid points can easily be obtained by transforming the metrics of the grid, which is numerically represented by the distance between the neighbour grid points.
A stochastic metrics, describing relative location uncertainties in the model operator $\mathcal{M}$, corresponds to the main effects that we want to simulate, because it can be thought to embed physical and numerical uncertainties that primarily affect the smallest scales. On the one hand, this can be viewed as an explicit transcription of Eq. (A6) in the internal model dynamics, and can thus be argued to describe uncertainties that upscale from unresolved processes. On the other hand, since the metrics is used everywhere in the model to evaluate differential and integral operators, making it stochastic can also be viewed as a simple
approach to simulate numerical uncertainties simultaneously in all model components.

In practice, to obtain a stochastic metrics in NEMO, we must transform the arrays describing the horizontal size of the grid cells into time-dependent stochastic processes. Thus, if $\Delta \mathbf{x}_i(t) = [\Delta x_i(t), \Delta y_i(t)]$ is the size of grid cell number $i$ at time $t$, we must define stochastic processes $\mathcal{P}_i$ such that:

$$\Delta \mathbf{x}_i(t + \Delta t) = \mathcal{P}_i \left[ \Delta \mathbf{x}_1(t), \ldots, \Delta \mathbf{x}_j(t), \ldots \right] \tag{A7}$$

A very simple approach to define the $\mathcal{P}_i$ is then to use first-order autoregressive processes $\boldsymbol{\xi}_i(t)$ as a multiplicative noise applied to the reference model grid $\Delta \mathbf{x}_i^0$:

$$\Delta \mathbf{x}_i(t) = \Delta \mathbf{x}_i^0 \circ [1 + \boldsymbol{\xi}_i(t)] \tag{A8}$$

with

$$\boldsymbol{\xi}_i(t + \Delta t) = \mathbf{a} \circ \boldsymbol{\xi}_i(t) + \mathbf{b} \circ \mathbf{w} \tag{A9}$$



where ∘ is the Hadamard product, $\mathbf{w}$ is a vector of independent Gaussian white noises, and $\mathbf{a}$ and $\mathbf{b}$ are constant coefficients governing the standard deviation and the correlation length scale of the $\boldsymbol{\xi}_i$. The components of $\boldsymbol{\xi}_i$ are thus assumed independent, which means that the grid is deformed independently along the two horizontal dimensions.

The use of autoregressive processes $\boldsymbol{\xi}_i(t)$ to simulate the stochastic distortion of the model grid makes the implementation of the scheme straightforward in NEMO, since we can directly apply the tools developed by Brankart et al. (2015) to generate the $\boldsymbol{\xi}_i$. This tool was indeed meant to be generic enough to trigger various sorts of stochastic parameterizations in NEMO, and has already been used to simulate various sources of uncertainty, including the effect of unresolved scales in the seawater equation of state (Brankart, 2013; Bessières et al., 2017; Zanna et al., 2019) and in the biogeochemichal equations (Garnier et al., 2016), or the effect of parameter uncertainties in the sea ice model (Brankart et al., 2015) and in the biogeochemichal model (Garnier et al., 2016). This tool only requires specifying a few parameters to characterize the stochastic processes $\boldsymbol{\xi}_i(t)$: the standard deviation ($\sigma$), the correlation time scale ($\tau$), the number of passes ($P$) of a Laplacian filter applied to the $\boldsymbol{\xi}_i$, and the order ($n$) of the autoregressive processes. The two last parameters go beyond the formulation of Eq. (A9), which describes first order processes (AR1) uncorrelated in space. The application of a Laplacian filter introduces space correlation and makes the distortion of the grid smoother in space, and the use of ARn rather than AR1 processes modifies the time correlation structure and makes the distortion of the grid smoother in time. It must also be noted that the use of ARn processes is also more general than Eq. (A7) by making the processes $\mathcal{P}_i$ depend on the $n$ previous time steps, rather than just the previous time step.

In the present study, the distortion of the grid has been limited to horizontal displacements of the model grid points, with the same displacements applied to all model fields and along the vertical. This reduces the number of stochastic fields to generate to two two-dimensional fields, one for each of the horizontal coordinates $\Delta x_i(t)$ and $\Delta y_i(t)$. However, since the NEMO fields are shifted according to the rules of the Arakawa C-grid, the stochastic metrics is first computed for the T-grid and then transformed to the other grids to be consistent with the shifted position of the grid points. In the application, the standard deviation is set to a relatively small value $\sigma = 1\%$ or 5%, to be consistent with the assumption of small location errors, and the correlation time scale is set to 1440 time steps (1 day) to be consistent with the assumption of a small variation of the grid over one time step. Some effort is also made to keep the perturbation smooth in space and time by applying $P = 10$ passes of a Laplacian filter and by using second order autoregressive processes ($n = 2$).

*Author contributions.* JLS, PB, TP and JMB proposed the initial model study. SL set up and ran the model with some assistance and instructive discussions with AA, LB, JMB, JMM and QJ. SL and JMB designed and applied the methodology to analyse the model outputs. SL and JMB wrote the initial manuscript and all authors commented on and contributed to draft revisions.

*Acknowledgements.* This project has received funding from the European Union Horizon 2020 research and innovation programme under grant agreement No 821926 (H2020-IMMERSE, https://immerse-ocean.eu/). The work was performed using HPC resources from GENCI-IDRIS, France (Grant A008-0101279).





## Appendix: References

Berner J., T. Jung and T.N. Palmer, 2012: Systematic model error: the impact of increased horizontal resolution versus improved stochastic and deterministic parameterizations. *Journal of Climate*, **25**, 4946–4962.

Bessières L., S. Leroux, J.-M. Brankart, J.-M. Molines, M.-P. Moine, P.-A. Bouttier, T. Penduff, L. Terray, B. Barnier, and G. Sérazin, 2017: Development of a probabilistic ocean modelling system based on NEMO 3.5: application at eddying resolution. *Geoscientific Model Development*, **10(3)**, 1091–1106.

Berloff, P. S., McWilliams, J. C., 2002: Material Transport in Oceanic Gyres. Part II: Hierarchy of Stochastic Models. *Journal of Physical Oceanographyi*, **32 (3)**, 797–830.

Brankart J.-M., 2013: Impact of uncertainties in the horizontal density gradient upon low resolution global ocean modelling. *Ocean Modelling*, **66**, 64–76.

Brankart J.-M., G. Candille, F. Garnier, C. Calone, A. Melet, P.-A. Bouttier, P. Brasseur and J. Verron, 2015: A generic approach to explicit simulation of uncertainty in the NEMO ocean model, *Geoscientific Model Development*, **8**, 1285–1297.

Brasseur, P. and Blayo, E. and Verron, J., 1996: Predictability experiments in the North Atlantic Ocean: Outcome of a quasi-geostrophic model with assimilation of TOPEX/POSEIDON altimeter data. *Journal of Geophysical Research: Oceans*, **101, C6**, 14161-14173, https://doi.org/10.1029/96JC00665.

Brodeau, L., J. Le Sommer and A. Albert, 2020: Ocean-next/eNATL60: Material describing the set-up and the assessment of NEMO-eNATL60 simulations (Version v1). *Zenodo*: http://doi.org/10.5281/zenodo.4032732.

Buizza, R., M. Miller, and T. N. Palmer, 1999: Stochastic representation of model uncertainties in the ECMWF ensemble prediction system. *Quaterly Journal of the Royal Meteorological Society*, **125**, 2887–2908.

Candille G., and O. Talagrand, 2005: Evaluation of probabilistic prediction systems for a scalar variable. *Q*uart. J. Roy. Meteor. Soc., **131**, 2131–2150.

Candille G., C. Côté, P. L. Houtekamer, and G. Pellerin, 2007: Verification of an ensemble prediction system against observations. *M*on. Wea. Rev., **1**35, 2688–2699.

Candille G., J.-M. Brankart J and P. Brasseur, 2015: Assessment of an ensemble system that assimilates Jason-1/Envisat altimeter data in a probabilistic model of the North Atlantic ocean circulation. *Ocean Science*, **11**, 425–438.

Chapron B., P. Dérian, E. Mémin, V. Resseguieri, 2018: Large scale flows under location uncertainty: a consistent stochastic framework, *Quarterly Journal of the Royal Meteorological Society*, **144(710)**, 251–260.

Diaconescu E. P. and R. Laprise, 2012: Singular vectors in atmospheric sciences: A review. *Earth-Science Reviews*, **113(3–4)**, 161–175.

Durand, M., L. Fu, D. Lettenmaier, D. Alsdorf, E. Rodriguez, and D. Esteban-Fernandez, 2010: The surface water and ocean topography mission: Observing terrestrial surface water and oceanic submesoscale eddies. *Proceedings of the IEEE*, **98 (5)**, 766–779.

Evensen, G. 1994: Sequential data assimilation with a non linear quasigeostrophic model using Monte Carlo methods to forecast error statistics. *J. of Geophys. Res.*, **99(C5)**, 10143–10162.

Frederiksen, J., T. O'Kane, and M. Zidikheri, 2012: Stochastic subgrid parameterizations for atmospheric and oceanic flows. *Physica Scripta*, **85**, 068 202.

Fu, L.-L., and R. Ferrari, 2008: Observing oceanic submesoscale processes from space. *Eos, Transactions American Geophysical Union*, **89 (48)**, 488–488.




Garnier F., J.-M. Brankart, P. Brasseur and E. Cosme, 2016: Stochastic parameterizations of biogeochemical uncertainties in a 1/4° NEMO/PISCES model for probabilistic comparisons with ocean color data. *Journal of Marine Systems*, **155**, 59–72.

Germineaud, C., J.-M. Brankart, and P. Brasseur, 2019: An Ensemble-Based Probabilistic Score Approach to Compare Observation Scenarios: An Application to Biogeochemical-Argo Deployments. J. Atmos. Oceanic Technol., 36, 2307-2326.

Griffa, A., 1996. Applications of stochastic particle models to oceanographic prob- lems. In: Adler, R., Müller, P., Rozovskii, B. (Eds.), Stochastic modelling in physical oceanography. Birkhuser Boston., pp. 113–140.

Hawkins E., R.S. Smith, J.M. Gregory and D.A. Stainforth, 2016: Irreducible uncertainty in near-term climate projections. *Clim Dyn* **46**, 3807–3819.

Hersbach H., 2000: Decomposition of the continuous ranked probability score for ensemble prediction systems. *Wea.* Forecasting, **15**, 559–570.

Juricke, S., P. Lemke, R. Timmermann. and T. Rackow, 2013: Effects of stochastic ice strength perturbation on Arctic finite element sea ice modeling, *Journal of Climate*, **26**, 3785–3802.

Juricke, S, MacLeod D, Weisheimer A, Zanna L, Palmer TN. , 2018: Seasonal to annual ocean forecasting skill and the role of model and observational uncertainty, *Q J R Meteorol Soc.*, **144**, 1947–1964.

Kalnay E., 2003: Atmospheric Modeling, Data Assimilation and Predictability. Cambridge University Press, Cambridge.

Lacarra J. and O. Talagrand, 1988: Short-range evolution of small perturbations in a barotropic model, *Tellus*, **40A**, 81–95

Leroux S., T. Penduff, L. Bessières, J.-M. Molines, J.-M. Brankart, G. Sérazin, B. Barnier, and L. Terray, 2018: Intrinsic and Atmospherically Forced Variability of the AMOC: Insights from a Large-Ensemble Ocean Hindcast. *J. Climate*, **31**, 1183–1203.

Leutbecher M., Lock S., Ollinaho P., Lang S.T., Balsamo G., Bechtold P., Bonavita M., Christensen H.M., Diamantakis M., Dutra E., English S., Fisher M., Forbes R.M., Goddard J., Haiden T., Hogan R.J., Juricke S., Lawrence H., MacLeod D., Magnusson L., Malardel S., Massart S., Sandu I., Smolarkiewicz P.K., Subramanian A., Vitart F., Wedi N. and Weisheimer A., 2017: Stochastic representations of model uncertainties at ECMWF: state of the art and future vision. *Quarterly Journal of the Royal Meteorological Society*, **143**, 2315–2339.

Lorenz E.N., 1965: A study of the predictability of a 28-variable atmospheric model, *Tellus*, **17**, 321–333.

Lorenz E.N., 1982: Atmospheric predictability with a large numerical model, *Tellus*, **34**, 505–513.

Lyapunov A., 1992: The general problem of the stability of motion. *International Journal of Control*, **55:3**, 531–534.

Mémin E., 2014: Fluid flow dynamics under location uncertainty, *Geophysical and Astrophysical Fluid Dynamics*, **108(2)**, 119–146.

Morrow R., Fu L.-L., Ardhuin F., Benkiran M., Chapron B., Cosme E., d?Ovidio F., Farrar J. T., Gille S. T., Lapeyre G., Le Traon P.-Y., Pascual A., Ponte A., Qiu B., Rascle N., Ubelmann C., Wang J. and Zaron E. D., 2019: Global Observations of Fine-Scale Ocean Surface Topography With the Surface Water and Ocean Topography (SWOT) Mission, *Frontiers in Marine Science*, **6**, 232.

Palmer, T.N., 2002: The economic value of ensemble forecasts as a tool for risk assessment: From days to decades. *Q.J.R. Meteorol. Soc.*, **128**, 747–774.

Palmer T., G. Shutts, R. Hagedorn, F. Doblas-Reyes, T. Jung, and M. Leutbecher, 2005: Representing model uncertainty in weather and climate prediction. *Annu. Rev. Earth Planet. Sci.*, **33**, 163–193.

Palmer T. and R. Hagedorn (Eds.), 2006: Predictability of weather and climate. Cambridge University Press.

Sakov P., Counillon F., Bertino L., Lisæter K.A., Oke P.R. and Korablev, A., 2012: TOPAZ4: an ocean-sea ice data assimilation system for the North Atlantic and Arctic, *Ocean Science*, **8**, 633–656.

Toth Z. and E. Kalnay, 1993: Ensemble Forecasting at NMC: The Generation of Perturbations. *Bulletin of the American Meteorological Society*, **74(12)**, 2317–2330.



Williams P.D., N.J. Howe, J.M. Gregory, R.S. Smith, and M.M. Joshi, 2016: Improved climate simulations through a stochastic parametrization of ocean eddies. *Journal of Climate*, **29(24)**, 8763–8781.

Ying Y.K. and J.R. Maddison and J. Vanneste, 2019: Bayesian inference of ocean diffusivity from Lagrangian trajectory data. *Ocean Modelling*, **140**, 101401, https://doi.org/10.1016/j.ocemod.2019.101401.

Zanna L., J.-M. Brankart J.-M., M. Huber, S.Leroux, T. Penduff and P. D. Williams (2019): Uncertainty and Scale Interactions in Ocean
Ensembles: From Seasonal Forecasts to Multi-Decadal Climate Predictions. *Q J R Meteorol Soc.*, **145(1)**, 160–175.