# Peer review of "Ensemble quantification of short-term predictability of the ocean dynamics at kilometric-scale resolution: A Western Mediterranean test-case."

_Ocean Science, 2022_

## Author Response (AR1)

**Manuscript os-2022-11**
**Response to Anonymous Referee #1:**

We thank the Reviewer for his/her careful reading of our paper, and for his/her remarks that will help improve the clarity of the manuscript. We did our best to take them into account as explained below. Our replies and comments are given in blue, while the original comments from the Reviewer are in gray.

*General comments:*

*This interesting manuscript focuses on the predictability of small scales in realistic ocean models kept "on track" by data assimilation (although the manuscript does not contain assimilation results). In particular, it proposes a rather novel methodological approach to relate forecast uncertainties to initial uncertainties in the fields, and presents some results quite convincingly in the context of a particular experimental protocol based on a set of 2D "displacements". The topic is scientifically relevant and important and the scientific quality is good, but the focus, clarity and precision could sometimes be greatly improved. I have no reservations about the statistical/probabilistic methodologies implemented, and the results are valid and interesting, but I am not convinced of their generality given the particular experimental protocol (type of uncertainties considered, seemingly "fixed" scale,number of members, etc.): the limits of the ensemble generation approach, and thus thescope and validity domain of the results, should become more apparent. This manuscript should eventually be accepted for publication, but perhaps not quite in its present form.*

*Specific comments:*

*The style of the introductory and methodological sections is sometimes rather "literary" and "rhetorical", convoluted to the point of being imprecise (an example: see the comment "lines 56-68" below) -- the approach is often introduced by invoking much more general and theoretical concepts than necessary. On other occasions, the text does not contain enough information or loses the reader. I would recommend (1) adopting a much more "direct", "factual", "scientific" style throughout the text, and (2) improving precision and conciseness. For example, when describing a methodology, the description of what was done in practice could be presented first, accurately and completely (and not in three different places, such as the perturbation scheme in sub-sections 2.2 and 3.1 and Appendix A); then the validity and scope of the approach, including the wider context, can be discussed, not the other way around.*

→ Yes, we agree that, in some places, the text could have been made more concise. We have tried to simplify the text where it was possible without changing the meaning of our arguments. We also believe that positioning the paper in the broader context is important for the reader to understand the method that is presented. In particular, in the case of the description of the perturbation scheme, we have modified the presentation to improve the clarity, but we kept the technical description of the implementation in the appendix. Otherwise, the main text of the paper would be even more lengthy and difficult to read.

*However, as the ms. progresses, the style improves, especially in the description of the results, which is often adequate.*
*The definition of predictability scores (in particular CRPS and predictability diagrams), and the way in which statistical calculations are carried out using all members of the ensemble in turn as a reference (reminiscent of generalised cross-validation) are two aspects of the work that could be generalised to problems beyond the particular experimental protocol. I was particularly interested in the dispersion of the CRPS estimates across the 20 cases (Figures 7,8) --*
*I would be curious to know what they look like with only the reliability*
*CRPS component or only the resolution component (the latter possibly giving access to a form of feature-based predictability, i.e. based on whether a particular forecast eddy is present across the members). The decorrelation score is interesting and also seems to be quite general. The location score is of course more related to the particular type of uncertainties in the study.*

→ In our application, the verification data that are used in the CRPS come from one additional member of the same ensemble simulation. So, by construction, the ensemble is always perfectly reliable, and the reliability component of the CRPS score should be zero. In practice, numerically, it is non-zero because of the limited size of the ensemble, but it is much smaller than the resolution component, which is what matters in our application to measure predictability.

*While section 4 is solid, one should keep in mind that the predictability analyses are conducted within a very specific experimental protocol: that of pseudo-random perturbations based on 2D "displacement" at small scales (10 grid points), having a direct impact on horizontal advection and pressure gradient at these scales (and indirectly on other dynamical processes). (I know that "displacement" is probably not the right term as*

*you are perturbing the metrics of the model operator, not the grid, but you could give this word that definition in your ms). That's OK, but in retrospect I probably would have liked a more honest introduction and summary framing the study more clearly in the particular experimental protocol (e.g. as described in subsection 2.2 from line 118). Indeed, the results in Section 4 could be very different for other forms of uncertainty. The conclusion is not careful enough in this respect: its first sentence ("The overall aim of this study...") promises too much in relation to the very real and effective work that has been done.*

*In addition, a limitation of this work that is not mentioned in the conclusion is that the correlation scale of the displacements is set (if I understood correctly) at 10 grid points. So, if I understand correctly, this is a predictability analysis study for a 10 grid point noise. Would a smaller or larger scale noise behave in the same way? What about pseudorandom correlation scales? However, I'm not sure I understood correctly, since the conclusion quotes "10 km wavelength" and not "a scale of 10 grid points" -- which is quite confusing. Similarly, the tenfold use of a Laplacian filter is mentioned -- even more Confusion.*

→ Indeed, some confusion might have arised from the text. We have made efforts to improve it at the different places where this aspect was mentioned. The stochastic perturbation is applied on the model grid (~1.4 km), together with a laplacian filter (10 passes) to introduce spatial correlations with neighboring grid points. The ensemble spread progressively develops and cascades upscale as seen for exemple on the spectral metrics in Fig.6.
We tested different numbers of passes for the laplacian filter in the range 3 -10 (not shown), without much difference in the behavior of the stochastic perturbation.

*Twenty members is a small size for an ensemble, again a topic not addressed by the conclusion. It is not clear whether we should interpret the discussion in 3.2.3 as evidence that 20 members are "sufficient" for the subsequent predictability study? What about the representation of spatial covariances with 20 members? (These generally converge more slowly than the variances). Also, what is the impact of the ensemble mean, and is it taken into account?*

→ Yes, 20 members is usually considered a small size for an ensemble that must be used in data assimilation systems, because they need an accurate and reliable description of the covariance matrix, describing the statistical dependence between model variables (in particular between observed and unobserved variables). In our

application, the objective is to study the evolution of the spread of a given quantity in the ensemble. This quantity may be a model variable, a spectral amplitude or the location of a structure, but this is always just one variable taken from different members. None of the scores described in the paper depend on the ensemble covariance. This is why predictability studies can usually be based on smaller size ensembles as compared to assimilation systems. Nevertheless, it is true that the accuracy of the measure of the spread also depends on the size of the ensemble (but less problematically than correlations), and that this should have been discussed. For instance, with 20 members, the accuracy of the ensemble standard deviation as an approximation to the true standard deviation is about 16%. This is obviously not perfect but sufficient to draw meaningful conclusions. A few words have been added to the conclusion of the paper to discuss this limitation.:

> "Of course, the ensemble size can be a limitation of the accuracy of the conclusions. In our case, with m=20 members, we can expect a 16% accuracy (1/sqrt(2m)) on the ensemble standard deviation as an approximation to the true standard deviation, which is not perfect, but sufficient to draw meaningful conclusions."

*This is a scientific paper. Therefore, the emphasis on CMEMS, which is cited several times, and which also comes as the "last word" in the conclusion, seems out of place. Such a study is of interest to all ocean forecasting systems. If appropriate, CMEMS can be mentioned in the acknowledgements.*

→ We have now replaced most occurrences of 'CMEMS' by operational systems / operational centers.

***Individual comments:***

*lines 56-58: This appears as a purely rhetorical statement, but perhaps I did not understand what was meant. Models and assimilated observations have errors which do impact the forecasts, we know that. Also, how can model instabilities be used to produce a valuable forecast?*

→ Yes, we agree, the sentence was very unclear. It has been replaced by :

> "What matters to the application is then the possibility to produce a valuable forecast with the model that is used (i.e. with its shortcomings and uncertainties), and which may

be quite different from what could be obtained by a perfect deterministic model (as would be done in traditional predictability studies)."

*lines 62-63: "initial uncertainties because observation resources are limited": yes, but observations have errors too; and in an assimilating system initial errors are also due to the whole history of all types of errors up to then.*
→ Yes, agreed. We clarified with:
*"initial uncertainties because observation and assimilation resources are limited, and model uncertainties because model resources are limited."*

*The introduction has no references on probabilistic skill scores.*
→ Our method to quantify predictability could be applied to any kind of score, so we chose to introduce probabilistic scores (i.e. CRPS) in the section where we use them as an example application of our method (i.e. section 4.1 )

*line 98: "initiated" -> "initialised"*
→ Yes, corrected. Thanks.

*section 2.1: Which scales can be accurately modeled by MEDWEST60? It is important to have those in mind in relation with the perturbation scales which you will use. Also, in the Mediterranean the internal Rossby radii are quite small.*
→ The stochastic scheme used in this work is designed to introduce uncertainty at model-grid scale, with a correlation length scale of 10 grid points, i.e. about 14 km. Uncertainty is thus introduced in the 10-18 km range of the Rossby radius of deformation in the region (e.g. Escudier et al, 2016 , their Fig.5, https://doi.org/10.1002/2015JC011371), which is resolved by ~7 to 13 grid cells in our model. We have now tried to clarify the text on those aspects at the beginning of section 3.2.1.

*lines 109-110: "In this context..." -> "In a purely deterministic approach..." to improve clarity.*
→ Yes, we clarfied by replacing « in this context » by « *In a purely deterministic approach* ».

*But still, you are missing modelling errors here (parameterisation, numerical schemes, missing physics).*
→ Yes, we do not claim that we include all possible sources of model uncertainties.

*lines 148, 151, in Table 2, etc: "probabilistic model" -> "stochastic model"*

→ Fixed.

*line 154, legend of Figure 2, etc: "grid size", "size of the model grid" -> "grid spacing" or "mesh spacing". Also what is the distribution law used for the perturbations? (If a noncompact support law is used, did you use an upper bound for the displacement?)*

→ Ok. "grid size" has now been replaced in the text as you suggested. Gaussian distributions are used for the perturbations. Since the standard deviation is very small, no bounds were needed.

*lines 163-164: "It does rely...": I do not understand the sentence (you wrote the opposite two sentences before). Also: part of this paragraph is descriptive, and part is a discussion in anticipation for another discussion in chapter 4: it is not good to mix everything because you'll get the reader lost.*

→ We have now removed the sentence that was unclear.

*Table 2: I do not understand what "identical" initial conditions mean. I would have thought that the spun-up fields would be pseudorandomly displaced using the 20 samples of the displacement fields (for each of 1%, 5% stdev), hence yielding 20 *different* initial conditions across the ensembles.*

→ All 20 member of the ENS-1% and ENS-5% experiments are initialized from "perfect" initial conditions (the same exact ocean state for each member), taken from the spinup simulation (which is a single simulation without any stochastic perturbation). As soon as the ENS-1% or ENS-5% starts, the stochastic perturbation is applied (representing model error) and it makes the members diverge.

*lines 182-183: I am a bit confused. The "displacement" is variable, with stdev = 1%-5% of the mesh spacing, but the displacement correlation scale is fixed to exactly 10 gridpoints. Therefore I do not understand the words "on the order of".*

→ Yes, the correlation scale is fixed to 10 grid points. This has been rephrased to « with a correlation length scale of 10 grid points, i.e. about 14 km ».

*Figure 3: It might be interesting to have a zoomed version on the right (perhaps just for low wavenumbers) to be able to see something.*

→ The main point with this figure is to show that the perturbed and unperturbed simulations are nearly undistinguishable from a spectral point of view (meaning that the stochastic perturbation added in the perturbed simulation do not alter the simulation of the physical quantities - here the SSH wavenumber spectrum). We have now modified the text in subsection 3.2.1 to clarify the purpose of this figure.

*I did not have time for a full second reading and hence for further individual comments, Sorry.*

**Manuscript os-2022-11**
**Response to Anonymous Referee #2:**

We thank the reviewer for his/her careful reading of our paper, and for his/her appreciation of the work done in this paper. We did our best to take the reviewer's remarks into account as explained below. Our replies and comments are given in blue, while the original comments from the Reviewer are in gray.

*Review of Leroux et al. "Ensemble quantification of short-term predictability of the ocean dynamics at kilometric-scale resolution: A Western Mediterranean test-case."*
*The manuscript presents an analysis of an ensemble of ocean model simulations at very high resolution using a novel idea for intrinsic model errors based on concepts of location errors. The article uses very solid and interesting concepts and methodology and makes both a refreshing and useful contribution to the operational ocean forecasting community (where I belong).*
*The exploitation part of the research is very well developed and thoroughly explained, which will certainly help popularise probabilistic diagnostics into the oceanographic community, but is so extensive as to almost entirely eclipse the core stochastic model developments, which constitute the novel aspect of the paper. It is indeed seldom that one sees a theoretical advance (that from the papers from Mémin and Chapron) brought into a realistic ocean model, so it is of general interest to see for the first time the effects of the stochastic perturbations on the model solution.*

*However there is no discussion of the numerical effects of these perturbations and no visual from the perturbed model (illustrations are disappointingly always extracted from the CI control simulation without stochastic noise), leaving an uncomfortable impression that something is hidden from the Readers.*
→ The stochastic scheme used in this work is designed to introduce uncertainty at the model grid scale (plus smoothing by a 10-passes laplacian filter to introduce spatial correlations with a few neighboring grid points). This uncertainty is then expected to develop and cascade spontaneously toward larger scales, through the model dynamics. Thus the stochastic perturbation introduced for that purpose should alter as less as possible the behavior of the physical quantities simulated by the model. This is why, by design, there is almost no visual difference between field

snapshots from the unperturbed model and from the perturbed models. For the sake of brevity we had mainly provided illustrations from the unperturbed experiment, which we consider as our main experiment in this study.
But we acknowledge the interrogations that it might have induced and we have now added a new figure (figure 3 in the revised manuscript, showing snapshots of the SST and relative vorticity simulated by the unperturbed model and the 2 perturbed models). We have also modified the text to clarify these issues at the beginning of section 3.2 and 3.2.1:

> "The stochastic scheme used in this work is designed to introduce uncertainty at model-grid scale, with a correlation length scale of 10 grid points, i.e. about 14 km. This uncertainty is then expected to develop and cascade spontaneously toward  larger scales through  the model dynamics. The  design should be such that the introduced perturbation alters as less as possible  the behaviour of the physical quantities simulated by the model.  Figure \ref{fig.mapSST} illustrates that indeed the simulated fields in the perturbed model remain  nearly unaltered and undistinguishable from the same fields in the unperturbed model. Only in the zoomed snapshot of relative vorticity (i.e. taking the Laplacian of Sea Surface Height, thus emphasising gradients) from experiment ENS-5\% (Fig. \ref{fig.mapSST}f), some visual alterations starts to appear on the smallest scales (note that this is why we did not propose any additional experiment with a stronger perturbation  than 5\% in our study)."

We also propose  for the reviewer some additional movies of the evolution in time of the SST field and relative vorticity field from experiments ENS-CI and ENS-GSL15 (unperturbed, and perturbed with a 5% std of the stochastic scheme) over the 2 months of simulation. These movies are in open access on vimeo: https://vimeo.com/showcase/9695743.

*Another aspect that is not discussed is the somewhat binary response of the model to the amplitude of the stochastic noise. The 1% case corresponds to 15m/d displacements (according to my own back-of-the-envelope calculation) and is most often indistinguishable from the CI (0%) case. On the contrary, the 5% perturbations corresponds to 75 m/d, which also seems tiny, turns out completely different from the other two cases and generates kilometers of feature location uncertainties within one single day. What happens between 1 and 5% that causes such a binary response? I believe that tidal amplification is the culprit and suggest an additional experiment in the detailed comments below, where the stochastic noise is turned off in the model nesting zone. The doubts on the stochastic perturbation method do not impair the main findings of the paper, because the latter probably stand with the*

*CI control ensemble alone, and the diagnostic methods can be applied to any stochastic model, but there is a risk that the manuscript is used to advocate for a stochastic model perturbation method that it does not truly validate.*

→ We understand that the presentation of the results may give the impression that the response of the model to the perturbation is binary, with almost no effect with a 1% perturbation and a large effect with a 5% perturbation. But the response is actually gradual, and in both cases, the dynamical behavior of the model is about the same as in the unperturbed model. By looking at one single member (whatever model variable), it is hardly possible to say if it is a perturbed or an unperturbed member (except by looking specifically at very fine details or at the spectrum in the fine scales in the case of a 5% perturbation). It is only by looking at the spread of the ensemble (misfit between members) that the effect of the perturbations can be very clearly detected, which is precisely what we do in the paper. And this effect on the spread is again visually magnified by looking at location misfits, which is a very sensitive diagnostic. So the reason why the effect on location misfits is much larger with a 5% perturbation is just that the perturbation itself has a 5 times larger standard deviation.

*Another general remark about the use of the probabilistic diagnostics is that some of them can be generalised to deterministic forecasts under ergodicity assumption: spatially averaged statistics (CPRS, PSD) can be interpreted as expectations and could be applied to forecast systems that have invested in high model resolution rather than in ensembles.*

→ It is true that some ensemble statistics can be reached also from a deterministic simulation under ergodicity assumption. But  it is not clear to us how you could generalize our approach and our diagnostics to a deterministic *forecast:*  the time-lag after which, starting from a given initial uncertainty, the final forecast score is quantified, has to be the same in all the realizations.
In addition, this would also require assuming the stationarity of the predictability statistics, which is far from obvious in complex non-autonomous systems. The system can be more or less predictable depending on its current state or on the atmospheric forcing conditions. This can only be assessed using an ensemble approach.

*Overall the paper is very good and makes a very enjoyable read. I am impressed by the*

*enormous amount of thoughts and work that went into it. The structure, the style and the illustrations are all excellent, and will certainly make a splash in the operational community. So I recommend its publication after revisions that I would call "major" because of a possible problem in the implementation of the stochastic method.*

*The paper is maybe a little on the long side but I will suggest some reduction of the illustrations and point out a few repetitions in the text. Ideally the manuscript should be split into two separate papers, one demonstrating a new stochastic perturbation method and the other on the ensemble forecast diagnostics, but I will not insist on this if the authors can shed more lights on the stochastic perturbation method without adding pages of text.*

→ Thank you again for the appreciation of the work done in this paper. We hope that we have provided enough new material to convince the reader that we only apply very small perturbations to the model operator, which produce only little effect on the model behavior (even if these small perturbations induce a substantial effect on the ensemble spread). That is why we did not expand much the description of the stochastic effect in the model (which is barely visible by itself), but only in the description of the effect produced on predictability (which is non-negligible as compared to initial uncertainties).

**Detailed comments:**
*Title, abstract and introduction: no remark. All are representing well the actual contents of the paper.*

**Section 2**
*- Figure 1: Why do you need to define as many as 3 subregions?*
→ We agree that it might not be optimal for the sake of clear presentation, but we ended up with 3 defined subregions in this work for technical reasons. (a) is the largest squared region to apply the spectral analysis, (b) was meant to be a zoom to illustrate fine-scale features on the snapshots, and ( c) is a small (100x100 points) region without land to apply our example score on the location of the features. We have now added *"and used for various diagnostics or visualizations"* in the caption of Figure 1 to be more explicit.

*- Line 90: I understand that the eNAT60 configuration is not only a boundary condition but a baseline to which the different experiments should revert if there were no stochastic*

*perturbations at all. Please make it explicit and come back to it whenever the different experiments are compared to eNAT60.*

→ We have now added an explicit mention to the eNATL60 experiment in paragraph 3.2.1 where a comparison of the wave-number spectra are made.

*- indicate which method is used to impose lateral boundary conditions (the Flather conditions?).*

→ The Flow Relaxation Scheme ("frs") is used for baroclinic velocities and active tracers (simple relaxation of the model fields to externally-specified values over a 12 grid point zone next to the edge of the model domain). The "Flather" radiation scheme is used for sea-surface height and barotropic velocities (a radiation condition is applied
on the normal depth-mean transport across the open boundary).
We have now added these technical details as a note in Table 1.

*- Line 95-98: a) and c) are not strictly a "difference" and b) should not lead to any difference as long as the model is numerically stable. Please rephrase.*

→ We have now rephrased this sentence to avoid using "difference" although we do think it is important to mention those technical aspects for the sake of reproducibility:

> "Compared to eNATL60 which was forced at the lateral boundaries by the daily GLORYS reanalyse \ref{LELL21} and  an  additional tidal harmonic forcing from the FES2014 dataset \ref{LYAR20}, in MEDWEST60 we add no additional tidal forcing since  it is already explicitly part of the hourly boundary forcing taken from the eNATL60 outputs. The model time-step in MEDWEST60 is also   increased  by a factor 2 compared to eNATL60 (80 seconds in MEDWEST60 versus 40 seconds in eNATL60."

*- Line 114-119: This argument is contorted. Any intrinsic or extrinsic errors (in the vertical mixing or winds for example) may as well affect the smallest scales of the ocean, if they are set up to do so. It would clarify the argument if you state upfront that you consider location errors exclusively and that other types of errors can be added at will.*

→ Yes, indeed, other sources of errors can directly affect the small scales. We have modified the text of the paper to correct this point:

*"These uncertainties are likely to depend on many possible sources, by embedding for instance misrepresentations of the unresolved scales and approximations in the model numerics, but also many others. "*

*- Line 134: Indicate the physical scales of 1% and 5% with respect to the temporal autocorrelation: displacements of 15 m/d and 75 m/d respectively.*

→ Yes, this information is indeed very helpful. In view of the typical grid size (1.4 km in average) and the correlation timescale of the perturbations (1 day), the typical velocity of the grid points is indeed about 14 meters per day (for the 1% perturbation) and 70 meters per day (for the 5% perturbation) in the two horizontal directions. This has been added in the text of the paper.

*- Line 139: "quite consistent" does not sound too good. Can you recall which conclusion of Mémin (2014) is comforted by the present study?*

→ We agree that the reference to Mémin in this sentence leads to confusion. It has been removed. What we do in this paper is not equivalent to what is done in the work of Mémin (2014). We just say that we use a « similar approach ». So, none of the conclusions obtained by Mémin (2014) can be comforted by this study. As explained in the paper, in the work of Mémin (2014) the noise is introduced in the continuous equations (as a random Lagrangian displacement of the fluid parcels) to obtain modified Eulerian equations (with additonal terms accounting for the noise), while in our study, the noise is directly introduced in the discrete model by a perturbation of the grid. The underlying idea is the same but we do not claim that it is equivalent. In addition, in the work of Mémin (2014), the noise is assumed uncorrelated in time (Brownian motion) as a basic assumption, while we assumed a 1-day decorrelation time scale.

*Section 3*
*- L. 158: what does CI stand for in ENS-CI? Control Integration?*
→ It stands for 'Conditions Initiales' (i.e. the source of uncertainty in the experiment) as opposed to ENS-1% and ENS-5% where the source of uncertainty comes from the stochastic perturbation. We have now made it more explicit in subsection 3.1 of the manuscript.

*- Figure 2b indicates that even after Laplacian smoothing, the square model grid is*

*distorted and deviates from orthogonality, which may lead to numerical noise and eventually instabilities. The ROMS user community is advised to keep the grid cells orthogonality above 95% in practice, and especially at the lateral boundaries of the model, to avoid errors propagating inside the model grid. My recommendations would therefore be to dampen the model grid perturbations in the nesting zone of the model (in the first 5 or 10 grid cells) to avoid inconsistencies between the outer an inner model solutions, in particular the barotropic mode. I will come back to this at Figure 4.*

→ Yes, it is true that too much distortion of the model grid cell can deteriorate the accuracy of the numerical schemes. On the other hand, our scheme is also intended to describe uncertainties in the numerics and thus to produce some spread at the numerical level. One perspective of development to alleviate possible difficulties might be to re-interpolate the model solution on the reference grid every while, or even at every timestep.

*- Table 2: Define e1 and e2 in relation to the appendix.*
→ Ok we have now replaced e1,e2 by Delta x Delta y in the Table.

*- Figure 3 shows indistinguishable lines, and no indication of what is good or bad. You could either plot the difference of PSD from the eNATL60 reference or solely indicate the maximum difference in the text and skip the figure altogether. If you keep the figure, I recommend to remove the part for wavelength > 250km because of the small domain.*
→ It seems important to us to keep this Figure, as it shows from a spectral point of view that the perturbed and unperturbed simulations are undistinguishable (meaning that the stochastic perturbation added in the perturbed simulation do not alter the simulation of the physical quantities (here the SSH wavenumber spectrum). In fact it comes back to your previous comment saying that there was not enough comparison of the perturbed and unperturbed simulations (see our answer to this comment). We have now modified the text in subsection 3.2.1 to clarify the purpose of Fig.3 (spectra) and new Fig.3 (snapshot).

*- Figure 4 exhibits an oscillatory signal in the ensemble spread, whereas intuitively I expect the spread to grow monotonously. The oscillations are most visible in the 5% case but also in the 1% case. I also noted that the oscillations peak at the same time in the 1% and the 5% cases, about 4 times a day. Unless you have used the same random seed in*

*the 1% and the 5% case - which would be odd - the coherent oscillations indicate an amplified resonance of tidal signals, which brings me back to my previous remark about barotropic lateral boundary conditions: the nesting routines (radiation condition or Flather conditions, whichever you use) should allow tidal and other barotropic signals to be evacuated out of the domain, but if the perturbations make this boundary condition imprecise, the tides may be reflected at the lateral model boundary and resonate inside the nested model domain. I have a suspicion that this could be avoided if the perturbations were attenuated near the model boundaries (and maybe in shallow waters as well).*

→ Yes, it is difficult to exclude the possibility that spurious numerical effects due to grid distortions have some impact on the solution, but it is difficult to speculate on this without running specific test cases (that would require significant additional computing resources).

*- Line 209: This claim could be confirmed by a look at the accuracy numbers from the MED MFC QuID document on the Copernicus Marine website.*

→ We have not found any reference to which we could compare based on *hourly* SSH in the region. But instead we have directly computed the time Std from the hourly SSH outputs from the up-to-date CMEMS Mediterranean Forecasting System at 1/24° and including tides (https://doi.org/10.25423/CMCC/MEDSEA_ANALYSISFORECAST_PHY_006_013_EAS6) and we found values consistent with our study. An example of plot is provided below (time Std of the hourly SSH over feb-may 2022 from the above dataset). Maximum signals are locally ~ 10cm which is consistent with our experiments.

[Figure]

→ We have now modified the text as followed in section 3.2.2:

"Those values are close to typical deviation values of  hourly SSH  over time in the Mediterranean region found in the CMEMS Mediterranean Forecasting System \citep{CLEM21} at same period of year (not shown)."

*- Figure 5 makes a stunning impression, but is uninformative. I would have preferred to see the 5% case to have a visual impression of the effect of random perturbations (there are otherwise none in the whole paper).*

→ As discussed above already, the stochastic perturbation was designed so that there is no visual effect of the perturbation on the physical fields. The point of this figure was rather to show the divergence between 2 members of the same ensemble (here we chose ENS-CI for the sake of brevity). See attached two supplementary figures (FIG06new_ENS-1.pdf and FIG06new_ENS-5.pdf) illustrating how 2 members diverge in ensembles  ENS-1% and ENS-5%.

**Section 4**

*- L. 268-280 is a nice introduction of the ensemble diagnostics, but seem like a methodological overkill: the diagnostics are initially intended for location-dependent comparison to observations, but in the absence of observations like in the present study, some more basic diagnostics may be simpler to use than a cross-validation with each ensemble member. This is the case for the CRPS which is aggregated spatially for all*

*members to a single number and does not seem to add more information than a standard deviation. Please replace by the ensemble spread if this is a simpler diagnostics that provides the same insights.*

→ Our argument in the paper is that cross-validation is useful if the objective is to measure predictability by comparison of different indicators (which can be more or less complex) to a reference truth, and not only by the standard deviation of the ensemble spread. To obtain general conclusions, it is then necessary to use each ensemble member as the reference truth, hence the cross-validation algorithm.

As a first simple indicator, we could indeed have used the rms misfit with respect to the reference truth rather than the CRPS score. And, in this case, the result would indeed probably not have been very different from simply looking at the ensemble spread. But for more complex indicators, the cross-validation algorithm is usually needed.

In practice, computing the CRPS score is not more complicated or more expensive to compute than the rms misfit. It was used on purpose to illustrate the fact that, with the cross-validation algorithm, predictability can be evaluated using any type of score of practical interest to the user. The only thing that is needed is an operator to measure some kind of misfit between a forecast and a reference truth.

*- L298-299 are repeated in the figure caption.*
→ It is on purpose that the text is repeated in the caption, as we wish the captions to be as informative as possible, even for a Reader that would only browse quickly the text and focus mainly on the figures.

*- Figure 8. It would seem fair to mention that beyond 5 days of lead time, the 95% percentile is dependent on the model trajectory and does not make a robust statistic, a larger ensemble or a different perturbation method may improve that.*

→ With the cross-validation algorithm, the result does not depend on the model trajectory since every member is used successively as the reference truth. The accuracy is thus only limited by the size of the ensemble. When the spread becomes large, the error is also larger in amplitude, which explains the irregular behaviour that can be seen in the figures.

*- The small lines in Figure 10 are not very informative. The three figures could be compressed into one by showing the three 95% quantile only and plotting the differences from the initial CRPS.*

→ The green line (i.e. showing the initial score required to have a 95% probability that the final score is below a given value)  only gives an illustration of  how our probabilistic definition for predictability can be read and used for quantitative results. We think that it is worth comparing the full probability distribution from the 3 experiments in the Figure, and not just the example of application (the green line).

*- Section 4.2.1: I guess there are technical difficulties with the location score in the presence of islands or complex coastlines. This could be mentioned.*

→ Yes, we fully agree that the location score used in the paper is just a first simple approach to further illustrate the point that any score can be used to evaluate predictability. (For instance, here, it would not be possible to measure the ensemble spread, cross-validation is really needed.) As it is, it has many shortcomings and should clearly be generalized if it must be used in practical applications. This is now acknowledged in the manuscript.

*- Figure 11 (top against bottom) is nearly showing the same thing. You could remove the two lowermost panels by adding the 20 isolines in the top panels.*

→ We tried to follow the Reviewer's suggestion (see Figure below) by adding the quantiles as contours on top of the SSS field in shading. But  we think the resulting figure is less easy to understand than the initial figure, so in the end we prefer to keep the initial one in the revised manuscript.

[Figure]

*- L. 433: Why choose SSH this time?*

→ We choose SSH for this last example score because SSH is an observed quantity, and SSH spectral analysis in space domain is often applied in studies focusing on submesoscale-permitting realistic ocean models (e.g. Ushida 2022, Adjayi 2021). We think the kind of probabilistic approach and score we illustrate here might be of interest for a larger audience than just the operational modeling community. This is why we also discuss the potential relevance of this kind of predictability diagrams in the context of the future SWOT altimetry mission, at the end of section 4.3.3.

*- L. 460: scales above 150 km should be removed from the figure.*

→ We have now added some grey shading in all the spectral figures for scales that are not fully resolved within the considered region (lambda>L/2 where L is the size of the region and lambda the spatial scale) and we also added some comments in the captions and text.

*- L. 461: I would suspect that checkerboarding (numerical noise) would easily cause the correlation of small scales. Numerical noise is ubiquitous in all ocean models although viscosity makes it almost invisible. If the authors use a high-contrast colour scale (like "details" in Ncview), they would probably see some checkerboarding in the model output, which would inevitably appear coherent at the smallest wavelengths of the model output.*

→ Yes. We had mentioned the possibility for numerical truncation errors in the text. We have now generalized to "numerical noise".

*- Figure 18: Add the diagonal line for T=0.*
→ We have now added a diagonal line for R_0 = R_forecast in the Figure.

*- L. 485: The authors could indicate which SWOT revisit time would be necessary to maintain the small-scale structures (if the data assimilation were ideally good).*
→ We have now added a few lines in section 4.3.3:

> "With a perfect model and a very good assimilation system that would ensure an initial ratio R_0 close to 1 (say 0.9 for the sake of the numerical application here) the spectral coherence ratio R of the forecast after 5 days drops down to 0.5 for scales in the range 10-30 km, while it remains above 0.8 for scales in the range 60-100 km at same time-lag. Or to put it differently, if the target for the spectral decorrelation was to remain above R=0.5 for all scales in the range 10-100 km, then a revisit time of the satellite between 5 and 10 days would be necessary."

*Appendix A1:*
*- L. 553: "Anamorphic transformation" is a pleonasm.*
→ Yes, but it is commonly stated like this. We modified the text to avoid the pleonasm: *"is a transformation of the coordinates (anamorphosis)"*

*- L. 582: the link between the theoretical papers from Mémin and Chapron and this one is not obvious. How does the sigma value translate into the stochastic process P?*
*Appendix A2*
→ Yes, we agree that the connection is not direct. The point is that there is no equivalence and thus no direct correspondence to find with the work of Mémin/Chapron, only close similarities. In the theoretical papers of Mémin/Chapron (leading to a continuous Eulerian model formulation), the noise is assumed uncorrelated in time, but they have a general formulation for the spatial correlation structure. On the other hand, in our simple pragmatic implementation (directly introduced as a Lagrangian displacement of the grid in the discrete model), the noise is assumed correlated in space and time, but with a very simple assumption for the space/time correlation structure.
The text of the appendix has now been modified as:

" σ(x,t) dB is a stochastic process uncorrelated in time, but correlated in space, with a general formulation of the spatial correlation structure. "

*- L. 595 to 599: "can be thought", "can be be viewed" and "can be argued" make a very embarrassed logical chain to line 600, which I would promote upfront to motivate the Approach.*

→ Here, the point is that we do not want to reduce the interpretation of the scheme to numerical uncertainties (line 600), but also to physical uncertainties (unresolved scales). We have tried to simplify the text to improve the clarity of the argument. The text has now been modified as followed in the appendix

" A stochastic metrics, describing relative location uncertainties in the model operator M, corresponds to the main effects that we want to simulate, because it can represent both physical and numerical uncertainties. On the one hand, the stochastic metrics is an explicitly Lagrangian transcription of Eq. (A6) in the model dynamics, which describes physical uncertainties that upscale from unresolved processes. "

*- L. 610: Mention that a^2 + b^2 =1 to maintain the variance constant.*
→ Depending on the situation, the variance is not always expected to be constant.

*- L. 611: The "assumed independence" of the perturbation is later contradicted by the Laplacian filter in Line 620.*
→ No, the application of the Laplacian filter does not modify the independence between the x and y components of the noise.

*- L. 618: the citation to Garnier et al. (2016) is repeated.*
→ Yes, sorry, the repetition has been removed.

*- L. 620: does the Laplacian filter maintain the standard deviation?*
→ The Laplacian filter does not maintain the standard deviation but a correction factor is applied afterwards to restore the specified standard deviation. This is now explicitly stated in the text of the appendix.

*- L. 620: is the value of sigma linked to the sigma in Mémin/Chapron?*

→ No, there is no link with the notation used in Mémin/Chapron. Here, it is just the standard deviation of the noise. In Mémin/Chapron, it is something like a square root of the spatial covariance of the noise.

*- L. 629: Transformed to the other grids: do you mean a linear interpolation?*
→ Yes, this is done by linear interpolation. This is now explicitly stated. The T-points are moved by the noise, and the U-points, V-points etc, are moved accordingly.

*- L. 632: Only here is it possible for the reader to calculate the typical scale of the perturbations (about 15 m/day for 1%). This information is important to realise how much the model amplifies the location noise into location errors (roughly by a factor of 100 to 1000 in a single day, which is mind-boggling) and should be discussed in the main text.*
→ Yes we agree that the scale of the perturbation is very important, but it was already provided  in section 2.2 (where we gave the standard deviations 1% and 5%). The text has now also been improved by giving explicitly the typical grid velocity (as suggested above by the reviewer).

**Typos:**
———
- l. 133: remove the second "that". —>FIXED.
- L. 239: "characterizing" —>FIXED.
- L. 343 Fussy -> Fuzzy —>FIXED.
- Section 4.3.1: "pf" -> "of" —>FIXED.
- L. 531: Beying -> Beyond—>FIXED.

---

## Author Response (AR2)

**Manuscript os-2022-11**
**Response to the Reviewer's comments:**

We thank the Reviewer for reading the revised manuscript, for his/her interest for the work that has been done, and for accepting our answers to his questions about the perturbation scheme, which, as we understood, was covering the major concern of his/her first review. Please find below our answers to his/her second review below:.

—

**1/** We are sorry to see that some misunderstanding seems to remain about the objective of our paper. The Reviewer misinterprets them as: "(1) the introduction of a new perturbation method for ocean models and (2) the popularisation of ensemble forecast metrics based on cross-validation in operational oceanography."
 In fact, the objective of our paper, as stated in the introduction, is *"to evaluate, the predictability of the ocean fine scales in a high-resolution (kilometric scale) NEMO-based model"*. And the methodology proposed in our paper to reach this objective is to compute necessary conditions on initial accuracy and model accuracy to achieve a given forecast accuracy (as explained in the introduction and summarized in the conclusion).

So, of course, to reach this objective, we need to make specific assumptions (as would be necessary in any practical system) about the metrics that is used to evaluate the forecast accuracy (interpreted by the Reviewer as objective (2) of the paper) and about possible sources of irreducible model uncertainty (interpreted by the reviewer as objective (1) of the paper).
The limitation of the study resulting from these choices are already acknowledged in the paper:

> a. The fact that the choice of the metrics/score is specific is acknowledged at the beginning of section 4. Any practical application needs to define its own goal. Cross-validation is a powerful tool to produce a predictability score not depending on a particular reference trajectory, but it is presumably not the only possibility.

> b. In section 2.2, the model perturbation scheme is presented as a possible generic solution to simulate uncertainties upscaling from the smallest scales of the model. It is used in the paper to illustrate the possible impact of irreducible model uncertainties on predictability. The fact that the conclusions depend on this assumption is already acknowledged in the introduction with:
> *"However, it is important to keep in mind that these conclusions will depend on the assumption made to simulate uncertainties in the system. Although generic, and designed to trigger perturbations in the small scales, they are still an approximation and cannot be expected to account for the full diversity of uncertainties of different kinds propagating in real operational systems."* .The suggestion by the reviewer to use atmospheric data as a possible source of irreducible model error (thus affecting predictability, and not just operational forecast accuracy) is debatable.

—

**2/** The Reviewer also points out the "shortcoming in the experimental setup, vastly underestimating uncertainties from operational forecast systems".
 We would like to emphasize that, following the logic presented in the introduction, the purpose of the paper is to evaluate the predictability of the system and usually this question is addressed by computing a lower bound for the forecast uncertainty resulting from irreducible uncertainty in the system (i.e. a vanishingly small initial error and/or irreducible model uncertainties). In this paper, we suggest that an alternative approach to investigate predictability in realistic systems is to look for an *upper bound* for initial and/or irreducible model uncertainties, *to obtain* a targeted forecast accuracy. In this respect, the results

obtained in the paper are realistic. As acknowledged in the text, they are only specific to the particular choice of the metrics, and the particular choice of irreducible model uncertainties (which only decreases the upper bound obtained with initial uncertainty only). Our goal is clearly not to simulate the same kind of initial error as in operational forecasting systems, which depends on the observation and assimilation system. This is clearly out of the scope of this study, as we have explained in the last two paragraphs of the conclusion.

We have now tried to clarify the objectives by adding the following text in the introduction :

*"In other words, the objective of this paper is thus to compute an upper bound (or more generally, necessary conditions) for the initial uncertainties, in order to obtain a targeted forecast accuracy. We do so by using different types of metrics to quantify the forecast accuracy, in order to emphasize that the definition of this metrics is still a subjective choice, which depends on the goal of every particular application. The influence of one possible source of irreducible model uncertainty on this upper bound will also be illustrated."*

And : *"It should be emphasized that the goal of the present study remains to quantify the intrinsic predictability of the system (as defined by Lorenz, 1995) and should not be confused with that of quantifying the prediction skill of any given current operational forecasting system, that would then incorporate all sources of error, such as extrinsic errors that would result from coupling with the atmosphere, sea ice etc (e.g. Robinson et al., 2002). However, deriving predictability as an upper bound or 'necessary conditions', as it is proposed in the present case study, can provide useful guidance for the design of the future generations of operational systems that (...)"*

—

**Detailed comments :**

- l. 225: The comparison of ensemble STD at one given time (your figure 5) to time-averaged std from the CMEMS system is not relevant for the discussion at hand.

First, 2.5 cm is the ensemble spread after two months of simulations while Clementi et al. run 10 days forecasts only. The accuracy indicated by Clementi et al is 4.2 cm as analysis error (day one) for daily data in the region of interest (Table 4, Region 2 in Clementi et al. 2021). Since daily averaging reduces the spread, the ensemble setup seems really far below target, say, by a factor of 20 to 100 rather than "comparable". Adjust the text here and dependencies elsewhere to indicate that the experiments are underestimating the operational uncertainties but the methods remain applicable.

We agree that some confusion might arise from this line of text and from this comparison and we have now tried to clarify..

In short, Figure 5 is used to illustrate the spread growth in our 3 experiments, and one of our comment about the figure  is to say that the spread  (ensemble STD)  has reached saturation after 2 months, with an amplitude at saturation of about 2.5 cm in average over the domain, with local maxima of spread values are found around 10 cm.

This amplitude seems to be consistent with time standard deviation of hourly SSH timeseries from CMEMS analyses available from this dataset : https://doi.org/10.25423/CMCC/MEDSEA_ANALYSISFORECAST_PHY_006_013_EAS6), which is what we can expect if the ensemble members are fully decorrelated after 2 months (saturation of the spread).

In any case, we feel that it is not the main point of our study, and we agree that our initial comment in the text and the reference to (Clementi et al. , 2021) for the above dataset might have brought some confusion. So we propose to simplify by removing the confusing sentence "*Those values are close to typical deviation values of 235 hourly SSH over time in the Mediterranean region found in the CMEMS Mediterranean Forecasting System (Clementi et al. , 2021) at same period of year (not shown)."*

What is perhaps more important to clarify is that with FIG.5, we were not commenting on the amplitude of the initial error (as the Reviewer might have misunderstood?). It should be noted that given the objective of the paper (i.e study predictability),  it would be inappropriate to generate levels of initial errors that are comparable to what exists in currently-used  operationnal systems. In fact, for our predictability analysis, we consider each timestep in the ensemble experiment as a virtual start date (thus bearing some initial spread of various amplitude), and we analyze the forecast accuracy at a given time-lag, *relative* to each of these virtual start dates.  So in other words,   it is not just the small initial condition of the ensemble

simulations shown in Fig. 5  that is used as possible initial error in this study, but the all range of ensemble spread available over the two months of these experiments (see our explanation in section 3.1: *"Note that the choice is made to (…)"*).

By looking at the CRPS score for SSH in Fig. 9 (a,d,g) for instance, it appears that there are situations where the initial error is as large as 4cm. The fact that current operational systems are unable to produce a sufficiently low level of initial accuracy would just mean that they would be unable to reach the level of forecast accuracy that we tested. But again, this depends on the observation and assimilation systems, and it is out of the scope of this predictability study.

- Detailed comment on Figure 5 (previously Figure 4) even though there is no time for new simulations, the authors should at least confirm the tidal (or inertial or other) nature of the oscillations in the ensemble spread and speculate their origin. This can be done by eyeballing the timing of the maxima in the curve, no need for additional experiments.

We have now added this comment in the text: *"From the figure, it appears that the presence of model uncertainty is associated with some  oscillatory behavior in the ensemble spread evolution, at a period close to half a day, and with amplitude growing with the amplitude of the model error (barely visible in ENS-1\% but appearing more clearly in ENS-5\%). These oscillations might reflect some slight spurious numerical effects due to the horizontal grid distortions imposed in these experiments with the parametrization. The period close to tidal  or inertial period, might suggest some effect related to partial wave reflexion  in the buffer zone at the lateral boundaries of the domain, but further investigation would be needed to be able to conclude."*

Figure 8 (previously 7): The text only pleads very generally for using ensemble techniques over deterministic simulations in order to use the metrics, which has limited interest. Looking at the curves, my impression is that there is no use in integrating the ensemble past 20 days because the CPRS stops increasing. A reflexion on this point would be welcome.

We think that it is important that the experiment is long enough so that the ensemble spread has time to saturate. This depends on the metrics that is used. On the CRPS scores for SSH, we agree that the spread saturates quite quickly, but on the larger scales, the saturation needs more time to be reached. This is what is shown on Figure 18, where scales larger than L=40 km are not yet fully decorrelated between the ensemble members after 20 days  (mean coherence ratio above 0.5 for scales L>40 km).

On Figure 9 (previously 8): The authors have missed my point: the 95% probability isoline is very erratic when the CPRS scores are clustered in bulgy-looking scatterplots. This illustrate my general comment that the exercise is academic and that the readers should be cautious when the results are tightly linked to a given trajectory. These scatterplots would likely look more homogeneous if several sources of errors were considered, and the text should note the limitation by the experimental setup.

Yes, the accuracy of the results is limited by the size of the ensemble simulation that has been produced, especially where the spread of the scatterplot is large. We have now acknowledged this  in the text of the paper (end of subsection 4.1.3): « *(with imperfect accuracy where the spread is large, as a result of the limited size of the ensemble)* ».

- L. 506: The revisit time is given for a perfect model and an almost perfect initial state and is on the optimistic side. Indicate that the necessary revisit time should be shorter to suit today's forecasting systems.

Yes. We have now added "*(and even shorter with current imperfect models).*" at the end of the sentence.

- L. 521-525: The authors should summarise their recommendations for further - more realistic - experiments.

As mentioned above, the objective of this study is not to reproduce realistic initial and modelling error as they are in realistic operational systems. In the context of a predictability study, our experiments are actually quite realistic. The only two specific choices that can be adjusted are (i) the metrics that is used to quantify the forecast accuracy, and (ii) the type of irreducible model uncertainties that is accounted for (as a possible further limitation to standard predictability). Also see our previous responses above.